# Mixed responses to targeted therapy driven by chromosomal instability through p53 dysfunction and genome doubling

Sebastijan Hobor[1,122], Maise Al Bakir [1,122], Crispin T. Hiley[1,2,3,122], Marcin Skrzypski[1,2,3,4,122], Alexander M. Frankell [1,2], Bjorn Bakker[1,5], Thomas B. K. Watkins[1], Aleksandra Markovets[6], Jonathan R. Dry [7], Andrew P. Brown[7], Jasper van der Aart [8], Hilda van den Bos[5], Diana Spierings [5], Dahmane Oukrif[9], Marco Novelli[9], Turja Chakrabarti[10], Adam H. Rabinowitz[11], Laila Ait Hassou[12], Saskia Litière[13], D. Lucas Kerr [10], Lisa Tan[10], Gavin Kelly [13], David A. Moore [2,14], Matthew J. Renshaw [15], Subramanian Venkatesan[1], William Hill[1], Ariana Huebner [1,2,16], Carlos Martínez-Ruiz [2,16], James R. M. Black [2,16], Wei Wu [10], Mihaela Angelova [1], Nicholas McGranahan [2,16], Julian Downward [17], Juliann Chmielecki[7], Carl Barrett[7], Kevin Litchfield [1], Su Kit Chew[1,2], Collin M. Blakely [10], Elza C. de Bruin [8], Floris Foijer[5], Karen H. Vousden [18], Trever G. Bivona [10,19], TRACERx consortium*, Robert E. Hynds [1,2], Nnennaya Kanu[2,123] ✉, Simone Zaccaria [2,20,123] ✉, Eva Grönroos [1,123] ✉ & Charles Swanton [1,2,3,123] ✉

The phenomenon of mixed/heterogenous treatment responses to cancer therapies within an individual patient presents a challenging clinical scenario. Furthermore, the molecular basis of mixed intra-patient tumor responses remains unclear. Here, we show that patients with metastatic lung adenocarcinoma harbouring co-mutations of *EGFR* and *TP53*, are more likely to have mixed intra-patient tumor responses to EGFR tyrosine kinase inhibition (TKI), compared to those with an *EGFR* mutation alone. The combined presence of whole genome doubling (WGD) and *TP53* co-mutations leads to increased genome instability and genomic copy number aberrations in genes implicated in EGFR TKI resistance. Using mouse models and an in vitro isogenic *p53*-mutant model system, we provide evidence that WGD provides diverse routes to drug resistance by increasing the probability of acquiring copy-number gains or losses relative to non-WGD cells. These data provide a molecular basis for mixed tumor responses to targeted therapy, within an individual patient, with implications for therapeutic strategies.

Up to 50% of all never-smokers who develop lung adenocarcinoma (LUAD) harbor tumors with mutations in the epidermal growth factor receptor (*EGFR*)[1,2]. *EGFR* mutations are predominantly clonal, making this an optimal therapeutic target. Unfortunately, only a minority of patients have a lasting treatment benefit for more than two years[3–5]. The median progression-free survival for patients receiving EGFR tyrosine kinase inhibition (TKI) therapy with osimertinib is 18.9 months and 10.1 months for patients with *EGFR* mutation-positive metastatic

A full list of affiliations appears at the end of the paper. *A list of authors and their affiliations appears at the end of the paper. ✉e-mail: n.kanu@ucl.ac.uk; s.zaccaria@ucl.ac.uk; Eva.Gronroos@crick.ac.uk; Charles.Swanton@crick.ac.uk

non-small cell lung cancer (NSCLC) when receiving treatment in the first and second line, respectively[6,7]. Primary resistance, with no objective treatment response, is seen in 20% and 29% of patients receiving osimertinib as first- and second-line treatment, respectively[7]. Acquired resistance is defined clinically as systemic progression as measured by RECIST (Response Evaluation Criteria in Solid Tumors[8,9]) after a period of initial response to EGFR TKI therapy[10]. Common resistance mechanisms include secondary alterations in *EGFR* itself, such as the "gatekeeper" T790M mutation in response to erlotinib treatment[11], as well as TKI bypass via alternative signaling pathways and/or somatic copy-number alterations (SCNAs), including amplifications of genes such as *MET, ERBB2, KRAS, NRAS,* and *BRAF*[12]. However, in ~30% of patients, the mechanisms of acquired resistance remain unknown[13,14].

Mixed treatment responses, also known as heterogeneous responses, where responding and non-responding metastases are detected within the same patient, have been observed with varying frequency in many cancer types. For example, 11% of patients with *BRAF*[V600E]-mutant melanoma or thyroid cancer had mixed responses after treatment with a BRAF inhibitor[15], whereas 56% of patients with renal clear cell cancer treated with anti-angiogenic tyrosine kinase inhibitors displayed mixed treatment responses[16]. In the CAIRO I/II studies of patients with colorectal cancer and liver metastasis, a mixed response to therapy was associated with poorer survival outcome[17].

When measuring response to therapy using current clinical RECIST version 1.1 guidelines[9], the sum of the diameters of all target lesions is used[9]. Within a cohort of patients classified as having a stable disease or partial response to treatment, there will be some patients with mixed responses i.e., responding lesions and, at the same time, progression of other lesions. Current RECIST reporting criteria do not consider such discordant radiological responses[18], nor do they conform to the standard definitions of acquired resistance to therapy since both responding and resistant lesions occur within the same patient simultaneously.

Only a limited number of studies have explored the prevalence of mixed responses to TKI in LUAD[18,19], and to our knowledge, none have investigated the underlying mechanistic basis of acquired resistance in this context. Since a single progressing lesion might contribute to systemic re-seeding, disease progression, early treatment failure and drug discontinuation, understanding the mechanisms of clonal diversification and intra-patient mixed tumor response dynamics may improve patient screening and therapeutic strategies[20].

*TP53* is mutated in around 40% of all patients with NSCLC, and *TP53* pathway perturbations in *EGFR*-driven tumors are associated with shortened progression-free (PFS) and overall survival (OS), in the context of treatment with first-, second-, or third-generation EGFR inhibitors[20]. It was recently suggested that loss of TP53 function, together with other genetic events, facilitates the acquisition of EGFR TKI resistance mutations[21]. Mechanistically, *TP53* loss of function permits the tolerance of chromosomal instability (CIN) and is enriched in whole genome-doubled (WGD) tumors[22–24]. Moreover, studies have demonstrated that WGD results in rapid propagation of CIN and acquisition of SCNAs[25–28].

We hypothesized that *TP53* loss together with WGD permits the rapid onset of CIN and SCNA acquisition, leading to more diverse tumor genotypes and phenotypes, thereby contributing to the radiologically observed mixed tumor responses within patients with clonal actionable driver events. We investigated this hypothesis in multiple clinical cohorts of patients with clonal *EGFR*-activating mutations treated with EGFR TKI, in genetically engineered mouse models (GEMMs) driven by clonal *EGFR* activating mutations with or without *Trp53* loss, and in isogenic cell lines to examine mechanisms of resistance and cellular evolution under therapeutic pressure using functional models, whole-exome DNA sequencing, and single-cell DNA sequencing.

## Results

### Mixed clinical responses to TKI therapy are prevalent in *EGFR*-driven lung adenocarcinoma

There is limited information available on mixed responses to targeted therapies and cytotoxic chemotherapy in NSCLC as most studies only report data required to meet RECISTv1.1 criteria for response[18]. We used the Reiter and Vogelstein[15] defined response parameters to distinguish homogeneous from mixed tumor responses in both human and murine datasets. Unlike RECISTv1.1, which defines progressive disease as a 20% increase in total diameter calculated as the sum of the diameter of all measured lesions, Reiter et al. defined response in individual lesions. A lesion was considered to respond if it shrank by at least 30% in diameter and stable if it did not grow more than 10% or reduce in size by more than 30%. Progression was defined by at least a 10% increase in lesion diameter. A homogeneous objective response to therapy was defined as having at least one lesion with a greater than 30% reduction in size, in combination with other lesions being stable (less than 10% growth) or reducing in size. The appearance of new lesions was not considered in the Reiter et al. definition of a mixed response. However, in the context of lesions that meet the criteria for a radiological response, we included the appearance of one or more new lesions in the mixed response classification, even if all other lesions were responding. If lesions within the same patient were assigned to both the response and progression criteria or the patient developed a new lesion, the patient was classified as having a mixed response to therapy.

All assessments of response were performed at a single time point where imaging was available and, unless specified otherwise, were performed at the first response assessment following treatment (12 weeks post-treatment ± 2 weeks). As we assessed radiological response on a lesion-by-lesion basis early in the course of treatment, we were able to identify non-responding lesions before a patient's overall tumor response reached the clinical definition of acquired resistance[10]. Therefore, to distinguish between clinical definitions of primary and acquired resistance, designed to standardize criteria for clinical trial enrollment, we refer to the growth of an established lesion as the "development of resistance". Similarly, we apply the same nomenclature to our murine data and the genetic aberrations associated with the development of resistance[19].

These radiological response parameters were first applied to analyse the European Organization for Research and Treatment of Cancer (EORTC) RECIST database[19], which contains response assessments from patients in phase II and phase III clinical trials. The last imaging assessment occurring during the first 12 weeks of treatment was used for the analysis and compared to the baseline pretreatment imaging assessment. The dataset includes response data from 8,365 patients with lung cancer (NSCLC and SCLC). Of the 428 NSCLC patients treated with erlotinib, 237 patients had at least two target lesions as defined by RECISTv1.1. Of these patients, 31% (73/237) had at least one responding target lesion that reduced in size by 30% or more (Supplementary Fig. 1a). Within this group, 34% (25/73) had a mixed response to erlotinib treatment. The majority of patients within this group (21/25), had growth of at least one existing target lesion (Fig. 1a, b and Supplementary Table 1), whereas a minority of patients (4/25), had a mixed response to erlotinib due to the appearance of a new metastatic lesion or progression in a non-target lesion (Fig. 1b). To summarize, in all patients where a mixed response to erlotinib could be measured, i.e., two or more target lesions could be assessed, 25/237 patients had a mixed response.

Of the 1633 patients treated with cytotoxic chemotherapy, 1092 patients had at least two target lesions. Within this group, 64% (699/1092 patients), had at least one lesion that shrank by 30% or more. However, this does not imply a RECIST response, as the sum of the diameters of the target lesions may have decreased by less than 30%. Within the 699 patients with one responding target lesion, 24% had a

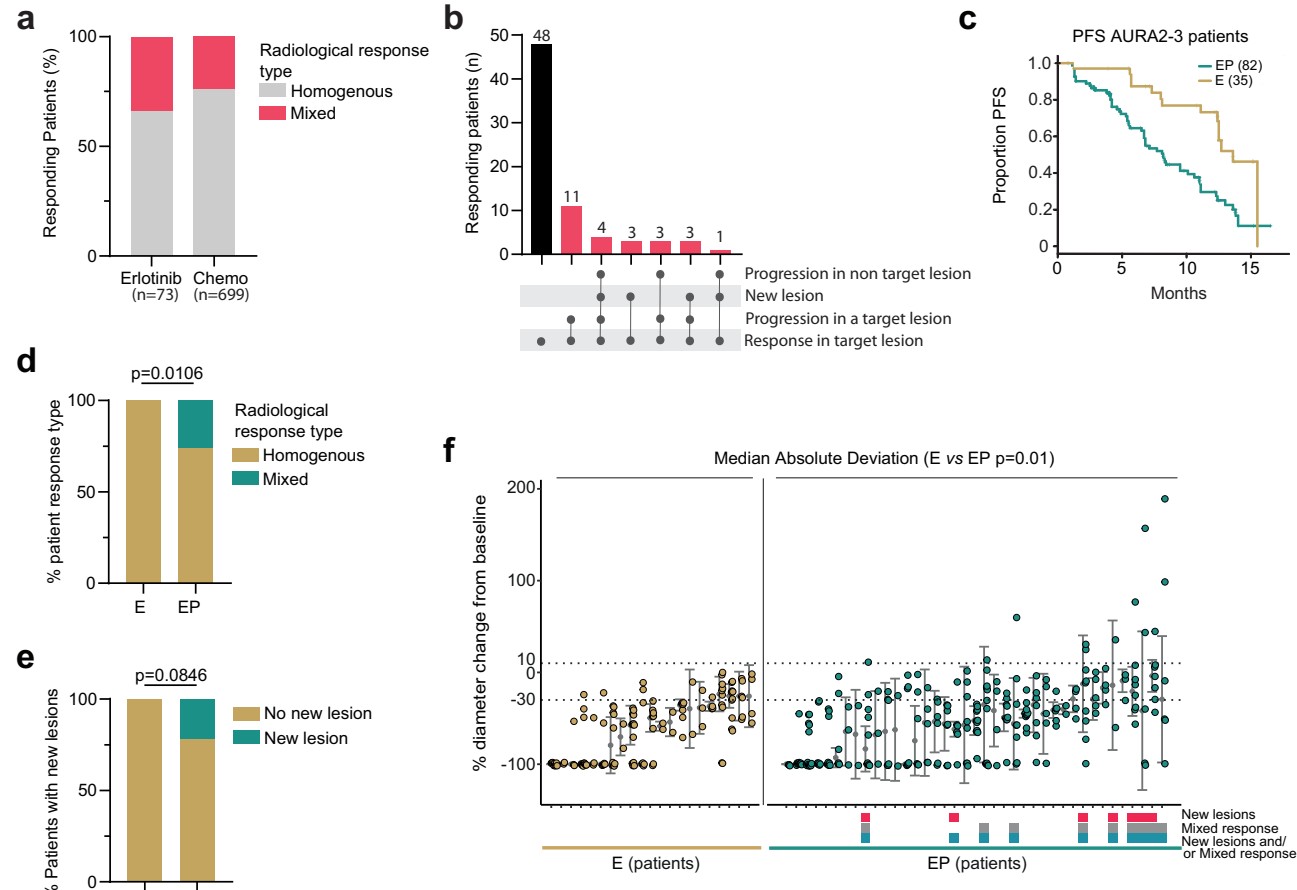

**Fig. 1 | TP53 pathway disruption is associated with shorter progression-free survival and mixed clinical responses to TKI therapy. a** Bar chart showing the percentage of responding patients with homogenous (gray) or mixed (red) responses to treatment with erlotinib or chemotherapy. **b** Mixed responses in the RECIST database were analysed using response criteria defined by ref. [15]. Patients with at least two lesions where one shrank by at least 30% were included in the analysis. The number of patients with homogenous responses are shown in black for patients receiving erlotinib. The different patterns of progression seen in patients with a mixed response are shown in red. **c** Kaplan–Meier survival analysis of patients with E ($n = 35$, yellow line) and EP tumors ($n = 82$, green line), in the AURA2, AURA3, and AURA phase II expansion cohort, demonstrating the difference in PFS after osimertinib treatment (Log-rank test (two-sided) $p = 4\mathrm{e}^{-04}$, HR 0.36, CI: 0.20–0.65). **d** Bar chart of the proportion of Homogenous (yellow) and Mixed (green) responses to osimertinib in patients with E or EP tumors ($p = 0.0106$

two-sided Fisher's exact test). **e** Bar chart of the proportion of patients with E or EP mutant tumours with new lesions during osimertinib treatment ($p = 0.0846$ two-sided Fisher's exact test). **f** Individual first tumor response within six months on osimertinib treatment, presented as % change in CT-measured tumor length. Each x-axis tick represents one patient ($n = 21$, E group of patients with 127 lesions and $n = 39$, EP group of patients with 246 lesions in total). The dotted lines show the Reiter et al. criteria for response (−30%) and progression (10%), respectively. Gray dots and whiskers represent the median change in tumor size and variability around the median value using the median absolute deviation (MAD) for each patient. Boxes underneath the graph indicate the occurrences of new lesions (red box), and mixed responses of existing lesions, as defined by ref. [15] (gray box), and patients with mixed responses with or without the occurrences of new lesions (blue box). Source data are provided as a Source Data file.

mixed response (165/699). Within this group of patients, a majority had growth of at least one existing target lesion as a component of the mixed response (106/165, Fig. 1a, Supplementary Fig. 1b, and Supplementary Table 2). In 59/165 patients, the mixed response was due to the appearance of a new metastatic lesion or progression in a non-target lesion. In all the patients where a mixed response to chemotherapy could be measured, i.e., two or more target lesions could be assessed, 165/1092 patients had a mixed response.

Focusing on those patients with at least one responding target lesion, we found that 34% (25/73) and 24% (165/699) of responding patients treated with erlotinib or chemotherapy respectively, displayed mixed responses to treatment (Fig. 1a). However, as responding patients were defined by the response in a single lesion rather than as the sum of diameters of the target lesions as per RECIST, our definition could include patients from across the RECIST response spectrum. Due to this reason, the response rates to erlotinib and chemotherapy reported here may differ from what would be expected clinically.

Indeed, using the response criteria outlined above, we found that within the EORTC cohort, 5.9% and 13% of erlotinib-treated patients, classified with partial response (PR) or stable disease (SD) according to the RECIST criteria, respectively, did in fact have mixed responses (Supplementary Fig. 1c, left panel). The equivalent numbers for patients treated with chemotherapy were 13.5% achieving SD and 16.9% for patients achieving a PR (Supplementary Fig. 1c, right panel). Although there are no scans or genomic data associated with the EORTC dataset, re-analysis of the underlying lesion measurements demonstrates that mixed responses are commonly observed in NSCLC patients treated with either TKI or chemotherapy.

## TP53 pathway disruption is associated with shorter progression-free survival and mixed clinical responses to TKI therapy

To understand the molecular basis underlying mixed responses in *EGFR*-mutant lung cancer, we used existing genomically annotated clinical cohorts. In clinical LUAD cohorts, loss of TP53 function has

been associated with reduced PFS and OS[20]. Molecularly, TP53 pathway disruption (defined as deleterious mutations in *TP53* affecting splice sites, DNA binding, transactivation domains and tetramer binding, *TP53* deletion, or clonal *MDM2/4* amplification) has been shown to increase the tolerance and propagation of genomic instability and CIN[29–32]. We hypothesized that TP53 pathway disruption in tumors with clonal *EGFR* activating mutations may promote cell-to-cell diversification by permitting the acquisition and propagation of SCNAs. An increase in SCNAs might be a substrate upon which selection for drug-resistant subclones could act and thereby expand phenotypic diversity and opportunities for the development of resistance, resulting in an increased frequency of mixed tumor responses within the same individual under the selective pressure of therapy.

The incidence of *EGFR* driver mutation-positive LUAD varies geographically and by genetic ancestry. In order to quantify the proportion of patients with *EGFR* driver mutation-positive LUAD with *TP53* co-mutation, we analysed the incidence of *EGFR* and *TP53* alterations in three well-annotated, geographically distinct LUAD cohorts. Concurrent TP53 pathway disruption was observed in 52% of the first prospectively recruited 421 patients in the TRACERx study (UK)[33], in 72% of the TCGA cohort (US)[1], and in 47% of the OncoSG cohort (East Asia)[34] (Supplementary Fig. 2a). Within the TRACERx421 cohort, there were 249 LUAD cancers sequenced, of which 25 harbored a clonal *EGFR* mutation. Of these 25 cases, seven were found to have clonal *TP53* mutations and two to have subclonal *TP53* mutations. An additional four tumors had clonal *MDM2/4* amplification events and the remaining 12 tumors were classified as *TP53* pathway wildtype (Supplementary Fig. 2b).

To investigate the effect of TP53 pathway disruption on targeted therapy response and TKI resistance, we used data from the AURA2 and AURA3 clinical trials, as well as the AURA trial phase II expansion cohort, which tested the efficacy of osimertinib in patients with metastatic *EGFR* mutation-positive NSCLC (Identifiers: NCT02094261, NCT02151981, and NCT01802632; see Supplementary Table 3 for patient selection criteria). All patients from the AURA clinical trials included in our analysis had NSCLCs that tested positive for both an *EGFR* activating mutation (e.g., L858R) and the EGFR[T790M] resistance mutation, had progressed following first-line EGFR TKI treatment and were treated with the third generation TKI osimertinib. Although some patients in these studies had an assessment of *TP53* co-mutation using circulating tumor DNA (ctDNA)[35], our PFS analysis of the combined AURA cohorts was restricted to patients with available tissue-based tumor somatic analysis ($n = 117$) due to the difficulty in calling copy-number loss from ctDNA and the confounding impact of clonal hematopoiesis of indeterminate potential on *TP53* mutations in circulating lymphocytes. Consistent with previous reports[20], patients whose tumors harbored oncogenic *EGFR* mutations and TP53 pathway disruption (EP, $n = 82$) had significantly worse PFS compared to patients whose tumors had only oncogenic *EGFR* mutations (E, $n = 35$), $p = 4e^{-04}$, HR 0.36, CI 0.20–0.65 (Fig. 1c). This difference could not be explained by a difference in the number of metastatic lesions present in the two patient groups before the start of osimertinib treatment (median numbers of lesions per patient: 5.5 E and 6 EP; $p = 0.7366$, Supplementary Fig. 2c, two-sided Mann-Whitney *U* test) or by other clinical variables (Supplementary Fig. 2d).

Next, response dynamics were examined in the 68 patients from the AURA cohorts who had at least two metastatic lesions at the baseline scan, had tissue-based tumor somatic analysis, and had consented to share longitudinal follow-up imaging (See Consort diagram, Supplementary Fig. 2e, for exclusion criteria). Within these trials, patients were imaged every 6 weeks following randomization. A total of 395 metastatic lesions from 46 patients with EP tumors and 22 patients with E tumors from the AURA2 ($n = 24$), AURA3 ($n = 33$), and AURA phase II expansion cohort ($n = 11$) were assessed to investigate homogenous and mixed responses during osimertinib therapy. In total, at the time of first assessment, 60/68 patients were defined as responders as they had at least one metastatic lesion that reduced in size by 30% or more (E: range 2–13 lesions per patient; EP: range 2–15 lesions per patient).

At the first follow-up time point, responding patients with EP tumors were significantly more likely to have mixed objective responses to therapy with the progression of one or more existing metastatic lesions consistent with the early development of resistance (Fig. 1d, 10/39 EP vs 0/21 E, $p = 0.0106$, Fisher's exact test). At this early time point, new metastatic lesions were only observed in the EP patient group, but this difference did not reach significance (Fig. 1e, 7/39 EP vs 0/21 E, $p = 0.0846$ Fisher's exact test). Patients with EP tumors had significantly higher variability in response between metastatic lesions (as measured using the median absolute deviation (MAD) of lesion response within each patient) and this was consistent whether measured at the first follow-up scan (Fig. 1f, $p = 0.01$ Mann-Whitney *U*-test) or at the point of maximum response where 64 out of 68 patients had at least one responding lesion (Supplementary Fig. 3a, $p = 0.032$ two-sided Mann-Whitney *U*-test). At the point of maximum response, patients with EP tumors continued to be more likely to have mixed responses to therapy (Supplementary Fig. 3b, 29/42 EP vs 5/22 E, $p = 0.0006$, two-sided Fisher's exact test) and were also more likely to progress with a new metastatic lesion (Supplementary Fig. 3c, 25/42 EP vs 5/22 E, $p = 0.0079$, two-sided Fisher's exact test) compared to patients with E tumors. No specific *TP53* mutation correlated with homogenous or mixed responses in this dataset (Supplementary Fig. 3d). We next investigated radiological responses from an additional dataset of osimertinib-treated patients from the University of California, San Francisco (UCSF) Clinical Cohort (Supplementary Fig. 3e). At the time of the first surveillance scan, new metastatic lesions were only evident in the EP group (0/14 E mutant group compared to 5/34 EP mutant group). We also analysed 113 lesions from the first surveillance scans of 31 patients (E = 9, EP = 22) with at least one responding lesion. Although we only could observe mixed responses in two EP patients at this early time point, the MAD of response was significantly different between the two patient groups (median of the MAD; E = 2% compared to EP = 13%, Supplementary Fig. 3f, $p = 0.0165$, Wilcoxon test). Focusing on patients with at least one responding lesion, there was a small but significant increase in the appearance of new metastatic lesions in the EP patient group when analysing the combined AURA and UCSF data-sets (0/30 E compared to 8/54 EP mutant group, $p = 0.049$) suggesting that loss of p53 function predicts the appearance of early new metastatic lesions in patients receiving EGFR TKI therapy. These data highlight the association between TP53 pathway dysfunction, mixed responses to EGFR TKI therapy, and reduced PFS in patients with metastatic *EGFR* driver mutation-positive NSCLC.

## *Trp53* loss is associated with more rapid therapy resistance and acquisition of alternative mechanisms of resistance to TKI therapy in mouse models

To model the impact of clonal p53 disruption in *EGFR*-mutant LUAD in the context of mixed responses to therapy, *EGFR*[L858R] (E) and *EGFR*[L858R]*Trp53*[fl/fl] (EP) GEMMs were used (Supplementary Fig. 4a). Lung specific expression and recombination was induced via intratracheal delivery of adenoviral Cre and tumor development was monitored using micro-CT scanning (See Supplementary Fig. 4b for workflow). EP mice demonstrated earlier tumor initiation (Supplementary Fig. 4c, $p < 0.0001$, two-sided Mann-Whitney *U*-test), a higher number of tumor nodules per mouse (Supplementary Fig. 4d, $p = 0.0124$, two-sided Mann-Whitney *U*-test), increased tumor proliferation indices (Supplementary Fig. 4e, $p < 0.0001$, Mann-Whitney *U*-test) and a trend towards higher tumor grade (Supplementary Fig. 4f, $p = 0.1704$, Kruskal-Wallis test) when compared to tumors from E mice. We observed significantly reduced overall survival of EP mice compared to E mice (Supplementary Fig. 4g, $p = 0.0014$, Mantel-Cox test).

We evaluated the suitability of GEMMs as a model system to assess mixed responses in *EGFR*-driven lung cancer by generating combined synteny SCNA maps of treatment-naïve mouse and human tumors. Re-mapping the mouse LUAD genome onto the human LUAD genome revealed that, in all samples investigated, oncogenes, such as *AKT1* and tumor suppressors, such as *PBRM1*, *SETD2*, *BAP1*, and *SMAD3*, were recurrently affected by copy-number gains (pink) and losses (blue) respectively in both human and mouse tumors irrespective of E or EP status (Fig. 2a). Syntenic gains or losses that were restricted to either E or EP tumors included the tumor suppressor gene *CDKN2A*, which was predominantly lost in mouse and human E tumors, whereas syntenic loss of *PTEN*, which has been associated with TKI resistance[36], was mainly observed in treatment-naïve EP tumors. These analyses demonstrate that tumors from E and EP mice recapitulate several of the genomic events observed in human tumors and highlight the potential importance of a limited set of genes commonly gained or lost in the earliest stages of *EGFR*-driven tumorigenesis (Fig. 2a and Supplementary Data 1).

Next, mice were scanned using micro-CT one month after Cre-mediated induction, and tumor development was monitored with monthly scans (see Methods). Erlotinib treatment was initiated upon identifying at least one lung tumor with a minimal diameter of 1 mm. If multiple smaller tumors were found (granular appearance of lungs), mice were scanned again after 2 weeks, and if it was deemed that the welfare of the animal would be compromised within the next 2 weeks, therapy was initiated (Supplementary Fig. 5a and Supplementary Data 2). As seen in treatment-naïve animals, EP mice had significantly reduced OS compared to E mice with treatment durations ranging from two weeks to 12 months (median survival after initiating erlotinib treatment: 13 (EP) and 34 (E) weeks respectively, Fig. 2b, $p < 0.0001$, log-rank Mantel–Cox test). Micro-CT imaging of response dynamics using the same parameters as the patient data in Fig. 1f, revealed that within the first month of therapy, tumors in individual E mice almost all uniformly responded to treatment with erlotinib (Fig. 2c). In contrast, EP mouse tumors exhibited significantly greater heterogeneity in response dynamics between tumors, with some lesions responding to treatment and others progressing within the same animal. To analyze the degree of variability in lesion size, the MAD percentage tumor diameter change within each animal was compared, which demonstrated significantly higher variability in tumor response in the EP than in the E mouse group (Fig. 2c, $p = 0.006464$, two-sided Mann–Whitney *U*-test, Effect size: −0.53232, Cohen's $d_s$).

The observation that acquired resistance was a rare event in the erlotinib-treated E mouse cohort but not in the EP mouse cohort (2 out of 12 E mice had at least one resistant lesion compared to 11 out of 16 EP mice, Fig. 2c) motivated us to further explore the association between TP53 pathway disruption and the development of resistance to therapy. The combination of longer latency times to tumor development together with fewer nodules per mouse in the E group (Supplementary Fig. 4c, d) prompted us to increase the probability of generating resistant tumors by adapting an intermittent dosing protocol shown to generate erlotinib-resistant tumors[37] (see Methods). Even when using this protocol, significantly more EP mice than E mice developed at least one therapy-resistant tumor (Fig. 2d, $p = 0.0082$, two-sided Chi-squared test). In addition, the development of resistance occurred significantly faster in EP mice (range 8–111 days after start of erlotinib) than in E mice (range 28–330 days after start of erlotinib, Fig. 2e, $p < 0.0001$, two-sided Mann–Whitney *U*-test). Within the first month of erlotinib treatment, 17/18 EP and 3/13 E mice had at least one resistant nodule, suggesting a higher propensity to develop early resistance to therapy in EP mice. Overall, these results indicate that Trp53 loss increases the probability of and reduces the time to developing therapy resistance.

To assess whether somatic resistance mutations, such as T790M, identified from repeat biopsy of patient tumors with clinically defined acquired resistance to TKI therapy[38], could explain the development of resistance in the mouse tumors, we performed whole-exome sequencing (WES, median depth of 92x, range: 58–169x) of 9 E and 10 EP erlotinib-resistant mouse tumors. There was no significant difference in the total tumor point mutational burden between the two genotypes in either the naïve or treated mouse tumor samples, or in the human TRACERx421, OncoSG, or TCGA cohorts (Supplementary Fig. 5b–e). Resistance to TKI treatment via histological transformation[13] of EGFR-driven LUAD to small-cell cancer with accompanying *RB1* mutations is a well-described phenomenon[39]. Neither *Rb1* mutations nor histological transformation was observed in the analysed resistant mouse tumor samples. In E-resistant mouse tumors, we identified four *EGFR* bypass mutations (oncogenic *Kras* mutations; Q61H, Q61R, and two G12D mutations[13]) and one gain-of-function mutation in *Fgfr2* (C382R)[40] (Fig. 2f). No acquired Trp53 mutations were observed in the E cohort after treatment. In contrast, only one known resistance-associated mutation, $EGFR^{T790M}$, was identified in an EP mouse tumor (Fig. 2f), suggesting that alternative mechanisms might be driving resistance in the remaining EP tumors.

## p53 dysfunction results in elevated SCNAs of genes associated with TKI resistance and increased cell-to-cell diversity

To investigate alternative, non-SNV-related mechanisms of resistance resulting from p53 loss of function, single-cell whole-genome sequencing (scWGS) was performed on FACS-sorted nuclei obtained from different ploidy groups within naïve and resistant E and EP mouse cells (see Methods). This approach revealed clear genomic differences between these four groups (Supplementary Fig. 6a, b). We utilized MEDICC to analyse the evolutionary timing of copy-number changes, including genome doubling, using the total copy number as input (see Supplementary Fig. 6c for workflow). A representative example of the derived phylogenies, gains, losses, and the timing of WGD is shown in Supplementary Fig. 6d. When investigating the earliest events on these phylogenetic trees (those within three edges from the most recent common ancestor, MRCA), we found a difference in the expansion of cells with copy number losses in E compared to EP tumors. In resistant E tumors, cells which underwent an early loss expanded very little, resulting in a lower cancer cell fraction (CCF) compared to EP cells with early losses (Fig. 3a, $p = 0.0435$, two-sided Mann–Whitney *U*-test). E cells with gains were more likely to expand and form the majority of the tumor. This phenomenon was not observed to the same degree in resistant EP tumors or in treatment-naïve tumors. This result is concordant with recent data obtained from cell lines, where TP53 loss correlated with a higher frequency of chromosome losses compared to isogenic TP53 WT cells[41]. The overall frequency of SCNAs across the genome was higher in the EP naïve mouse and human (OncoSG) tumors compared to E tumors (Supplementary Fig. 7a, b Mouse: median frequency E 15% vs EP 38%, $p < 2.2e{-}16$ Human: median frequency E 8% vs EP 12%, $p < 2.2e{-}16$; two-sided Mann–Whitney *U*-tests). Due to the relatively low number of patients with *EGFR* mutations in the Tx421 cohort, there was insufficient statistical power to perform this analysis in that dataset.

Further assessment of the extent of chromosomal alterations revealed that EP mouse tumors exhibited a higher frequency of both copy-number gains and losses across the whole genome compared to E tumors in both treatment naïve and resistant settings (Fig. 3b upper *vs* lower genome wide plots, Supplementary Fig. 7c, naïve E vs EP gains $p = 0.0002$, resistant E vs EP gains $p = 0.0013$, naïve E vs EP losses $p = 0.0244$, resistant E vs EP losses $p = 0.1679$, two-sided *t*-tests).

While no copy-number gains were observed to be significantly more frequent in E TKI-resistant mouse tumors compared to E treatment-naïve mouse tumors (Fig. 3b, upper panel), a higher frequency of copy-number gains of genes implicated in TKI resistance was observed in EP-resistant mouse tumors compared to EP treatment-naïve tumors (Fig. 3b, lower panel). For example, a region of

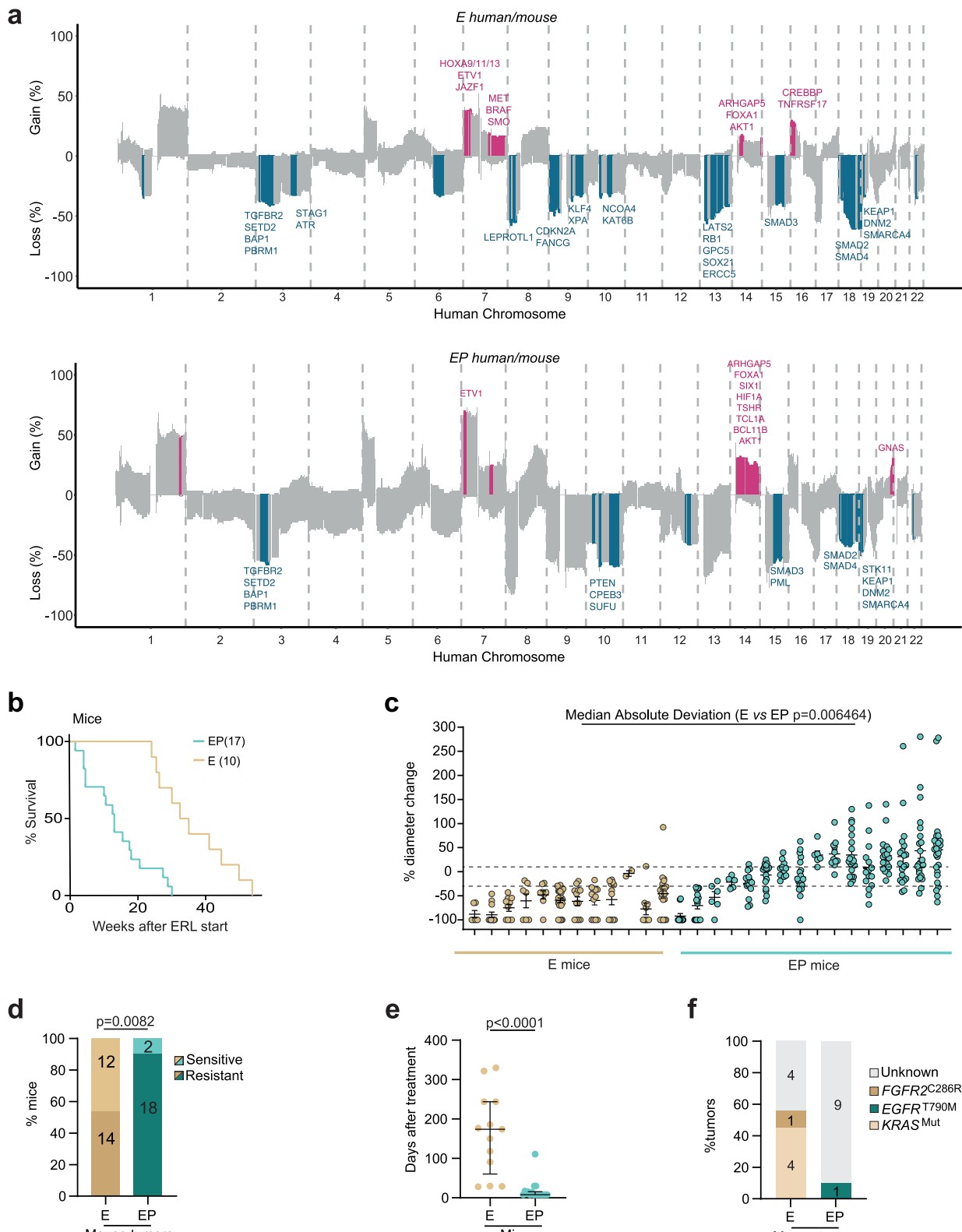

**Fig. 2 | Trp53 loss results in mixed responses and therapy resistance in murine models of NSCLC. a** Mouse-to-human across genome synteny histograms. Upper panel E mice vs. patients with E tumors. Lower panel EP mice vs patients with EP tumors. Significantly changed regions in both species are colored pink (gain) and blue (loss). **b** Kaplan–Meier survival analysis of E ($n = 10$, yellow line) and EP ($n = 17$, green line) mice, demonstrating the difference in OS after erlotinib treatment ($p < 0.0001$, HR 3.72, 95% CI: 1.65–8.38, log-rank Mantel–Cox test). **c** Differences in tumor responses after one month of erlotinib treatment in E ($n = 12$ yellow) and EP ($n = 16$ green) mice, presented as % change in CT-measured tumor diameter. Each column represents one mouse, and each dot represents one tumor within the mouse ($p = 0.006464$, two-sided Mann–Whitney $U$-test). The dotted lines show the Reiter et al criteria for response (−30%) and progression (10%), respectively. **d** Bar chart showing the proportion of sensitive and resistant tumors in E (yellow) and EP (green) mice ($p = 0.0082$, two-sided chi-squared test). The total number of mice in each group are indicated in the bars. **e** Dot plot showing time to resistance in E ($n = 13$ yellow) and EP ($n = 18$ green) mice ($P = <0.0001$ two-sided Mann–Whitney $U$-test). **f** Bar chart showing identified single-nucleotide variant-related resistance mechanisms in E and EP mice. Source data are provided as a Source Data file.

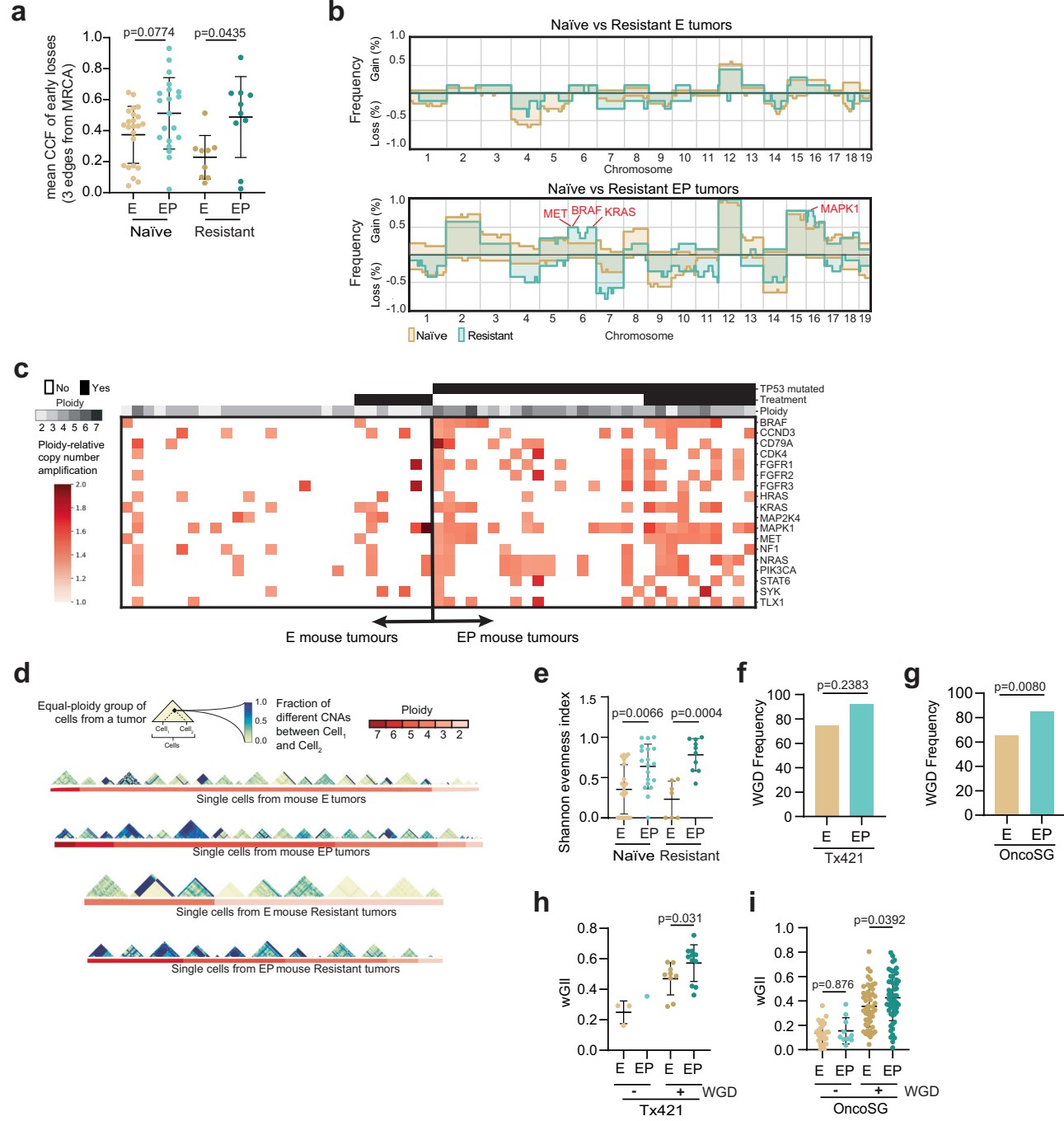

**Fig. 3 | Trp53, together with WGD, results in therapy resistance associated with increased CIN and cell-to-cell variability. a** Dot plot of mean cancer cell fraction per tumor of early losses in naïve (E, $n = 23$; EP $n = 20$) and resistant E ($n = 9$ yellow) and EP ($n = 10$ green) tumors. **b** Frequency of copy-number gains (positive y-axis) and losses (negative y-axis) in treatment naïve (yellow) vs resistant (green) tumors with either E (upper panel) or EP (lower panel) genotypes. **c** Ploidy-relative copy-number gains (red colors) are reported across all mouse tumors, separating treatment naïve vs resistant and E vs EP (ploidy represented in gray colors) for 18 genes whose amplification is known to have an impact on TKI resistance. **d** For every group (triangle shape) of single cells obtained from the same FACS ploidy peak (red colors) from either naïve E (top row), naïve EP (second row), resistant E (third row), or resistant EP (bottom row) mouse tumors, the fraction of the genome affected by different SCNAs (yellow-to-blue colors) was computed between every

pair of cells (square within a triangle) as a proxy to measure cell-to-cell diversity. **e** Dot plot of average Shannon evenness index measured per tumor from naïve ($n = 21$) and resistant ($n = 7$) E (yellow) vs naïve ($n = 19$) and resistant ($n = 710$) EP (green) mouse tumors (naïve $p = 0.0066$, resistant $p = 0.0004$, two-sided Mann–Whitney $U$-tests). Bar charts showing WGD frequencies in lesions from patients with E and EP tumors from **f** Tx421 (using WES data, ns, two-sided chi-squared test) and **g** OncoSG cohort (using WES data, $p = 0.0080$, two-sided chi-squared test). Dot plot showing weighted Genome Instability Index (wGII) of tumors with or without WGD in patients with E or EP lesions from the **h** Tx421 (non-WGD: E $n = 3$, EP $n = 1$. WGD: E $n = 9$, EP $n = 11$ $p = 0.0310$, two-sided Mann–Whitney $U$-test) and **i** OncoSG cohorts (non-WGD: E $n = 26$, EP $n = 10$, $p = 0.8758$. WGD: $n = 9$ E, $n = 11$ EP, $p = 0.0392$, two-sided Mann–Whitney $U$-test). Source data are provided as a Source Data file.

chromosome 6p, harboring *MET* and *BRAF*, was significantly more frequently gained in EP resistant compared to EP treatment-naïve tumors (Fig. 3b lower panel, $p = 0.01135$, see Methods). Similarly, genomic regions in chromosomes 6q and 16p that, amongst others, encode genes associated with EGFR TKI resistance, such as *Kras* and *Mapk1*[13], were more significantly gained in EP TKI-resistant mouse tumors compared to EP treatment-naïve mouse tumors (Fig. 3b lower panel, $p = 0.0007$, see Methods).

Based on these data, we analysed the copy-number status of several additional genes that have been implicated in TKI resistance[12–14,42] and investigated the SCNAs between the two genotypes. A significantly higher frequency of copy-number gains involving published TKI resistance associated genes was observed in EP treatment-naïve mouse tumors compared to E treatment-naïve tumors (Fig. 3c and Supplementary Fig. 7d left panel $p = 5e^{-10}$, Chi-squared test), as well as in EP-resistant mouse tumors compared to E resistant mouse tumors (Fig. 3c and Supplementary Fig. 7d right panel, $p = 2.7e^{-6}$, two-sided Chi-squared test), which may contribute to the development of resistance.

To investigate the extent of heterogeneity within the naïve and resistant E and EP tumors, the SCNAs in individual cells were inferred from scWGS data, and cell-to-cell diversity was estimated by measuring the fraction of the genome affected by different SCNAs between every pair of cells obtained from the same tumor and the same group of cells with the same ploidy (Fig. 3d, Methods). Consistent with a role for TP53 pathway disruption in the potentiation of CIN, single cells from both naïve and resistant mouse EP tumors were found to display significantly higher genome ploidies than E tumors (Supplementary Fig. 7e) and a higher prevalence of WGD in resistant tumors (naïve E vs. EP; $p = 0.1365$, resistant E vs. EP; $p = 0.0060$ two-sided Chi-squared test, Supplementary Fig. 7f). A higher extent of cell-to-cell diversity was also observed between cells derived from EP tumors compared to single cells from E tumors, which resulted in significantly higher intra-tumor heterogeneity (as measured by Shannon evenness index, Fig. 3e, naïve $p = 0.0066$ and resistant, $p = 0.0004$ tumors, two-sided Mann–Whitney $U$-tests). The cell-to-cell diversity observed in the EP tumors reflected a significantly higher weighted genome instability index (wGII), consistent with the greater chromosomal complexity and elevated burden of SCNAs[43] in both naïve and resistant EP mouse tumors compared to E mouse tumors (Supplementary Fig. 7g, h; naïve $p < 2.22e^{-16}$ and resistant $p < 2.22e^{-16}$, two sided Mann–Whitney $U$-tests). Taken together, these results suggest that the development of resistance in E tumors is often driven by point mutations (Fig. 2f), whereas both human and mouse EP tumors have greater SCNA heterogeneity leading to the selection of SCNAs encoding genes known to drive resistance to EGFR TKIs.

To ascertain the contribution of WGD to the elevated genome instability in EP tumors, we next examined WGD events in the Tx421 and OncoSG cohorts and observed that WGD tended to be more frequent in EP tumors than in E tumors (WGD frequency; Tx421 71.4% (E); 85.7% (EP); OncoSG 65.3% (E); 84.8% (EP); ($p = 0.2383$, Fig. 3f and $p = 0.008$ Fig. 3g two-sided Chi-squared tests). However, these small differences in WGD frequencies alone were unlikely to explain the profound phenotypic differences in resistance dynamics between E and EP tumors, prompting us to investigate whether WGD could be associated with elevated CIN in a manner dependent on p53 pathway dysfunction. When further assessing the extent of genome instability, as measured by the wGII, we observed higher wGII in WGD EP tumors compared to WGD E tumors in the Tx421 and OncoSG cohort (Fig. 3h, i; Tx421; $p = 0.031$, OncoSG; $p = 0.0392$, two-sided Mann–Whitney $U$-tests) suggesting that WGD is associated with elevated CIN, which is more pronounced on a *p53* mutant background. This observation was also recapitulated when assessing the effect of WGD on genome instability in *KRAS* and *KRAS*/p53 pathway mutant tumors in the Tx421 cohort (Supplementary Fig. 7i, $p = 0.0004$, two-sided Mann–Whitney $U$-test).

## Combined WGD and the presence of p53 dysfunction generates a permissive landscape facilitating genetic resistance to TKI

Based on these data, we hypothesized that WGD, together with p53 dysfunction, accelerates cell-to-cell variation in the acquisition of SCNAs, generating a diversity upon which selection can act. In the context of EGFR TKI therapy, this genomic diversity may promote more rapid acquisition of resistance and mixed responses seen in the human and murine data compared to E tumors with functional p53.

To decipher the contribution of WGD in acquired drug resistance on a background of p53 dysfunction, we used an isogenic clonal *EGFR/TP53* mutant human NSCLC PC9 cell model system, with and without an additional WGD event. The LUAD cell line PC9 is triploid and harbors both an oncogenic *EGFR Ex19del* and an inactivating *TP53* mutation (p.Arg248Gln)[11,44–47]. Similar to other cancer cell lines[26,48], a small fraction of PC9 cells undergo spontaneous WGD events in cell culture. Using single-cell sorting, we obtained cells with a relative DNA content of 3N (triploid, 24 single-cell clones isolated) and isogenic cells that had spontaneously undergone an additional WGD event with a relative DNA content of 6N within the parental 3N population (hexaploid, 24 single-cell clones isolated; Supplementary Fig. 8a, b).

A subpopulation of PC9 cells harbors the *EGFR* T790M mutation, which is the most frequent resistance mechanism in response to erlotinib treatment[11,45,49]. However, all 24 triploid and 24 hexaploid-derived early passage WGD cell populations were equally sensitive to erlotinib (Supplementary Fig. 8c), indicating that T790M mutations were absent from the clonal founder cells and that at baseline, these clones had not acquired additional genetic alterations associated with *EGFR* TKI resistance. Importantly, the IC50 values of these early passage triploids (T) and hexaploid (H) progenitor clones were comparable to the parental population (IC50 ≈ 15 nM), confirming that a spontaneous WGD event alone on a p53 mutant background does not confer drug resistance (Supplementary Fig. 8c). To investigate the emergence and frequency of resistance, each of the 48 progenitor clones were seeded into a full 96-well plate each (5000 cells per well) and cultured in the presence of 1.5 μM erlotinib (we defined acquired resistance as survival in a 100-fold higher concentration than the observed median IC50 value of the parental clones[50], see Supplementary Fig. 8a, right panel, for workflow). The emergence of resistant subclones was recorded after 5 weeks of continuous erlotinib treatment. In parallel, each progenitor clone was cultured in the absence of erlotinib to investigate a potential drift in copy-number status (Supplementary Fig. 8d, see below).

Both triploid and hexaploid clones were able to generate erlotinib-resistant subclones within the 5-week time period. A significantly higher proportion of hexaploid progenitor clones generated at least one resistant subclone compared to their triploid counterparts, indicating that within five weeks of drug exposure, a WGD event together with p53 dysfunction promotes the development of drug resistance (12/24 triploid vs 19/24 hexaploid progenitors, $p = 0.0346$, Chi-squared test, Fig. 4a, b). To investigate a mechanistic basis for this observation, we performed WES of 34 triploid- and 40 hexaploid-derived erlotinib-resistant subclones. Assessing the presence of known resistance mechanisms across the subclones, we found that 15/34 triploid-resistant subclones harbored a T790M mutation and 11/34 triploid-resistant clones harbored a RAS/PI3K pathway activating mutation. In contrast, 19/40 resistant hexaploid/genome-doubled subclones harbored mutations in T790M (15/40) or the RAS/PI3K pathway (4/40), with the remaining clones having no known point mutation mechanism of resistance; Fig. 4c, left panel). In line with a recent publication[51], different routes to resistance were observed in daughter clones derived from a single parent clone. We also observed variability in the ploidy of daughter clones derived from the same parental clone. Taken together, triploid-resistant subclones (26/34) were significantly more likely to harbor an SNV as a mechanism of

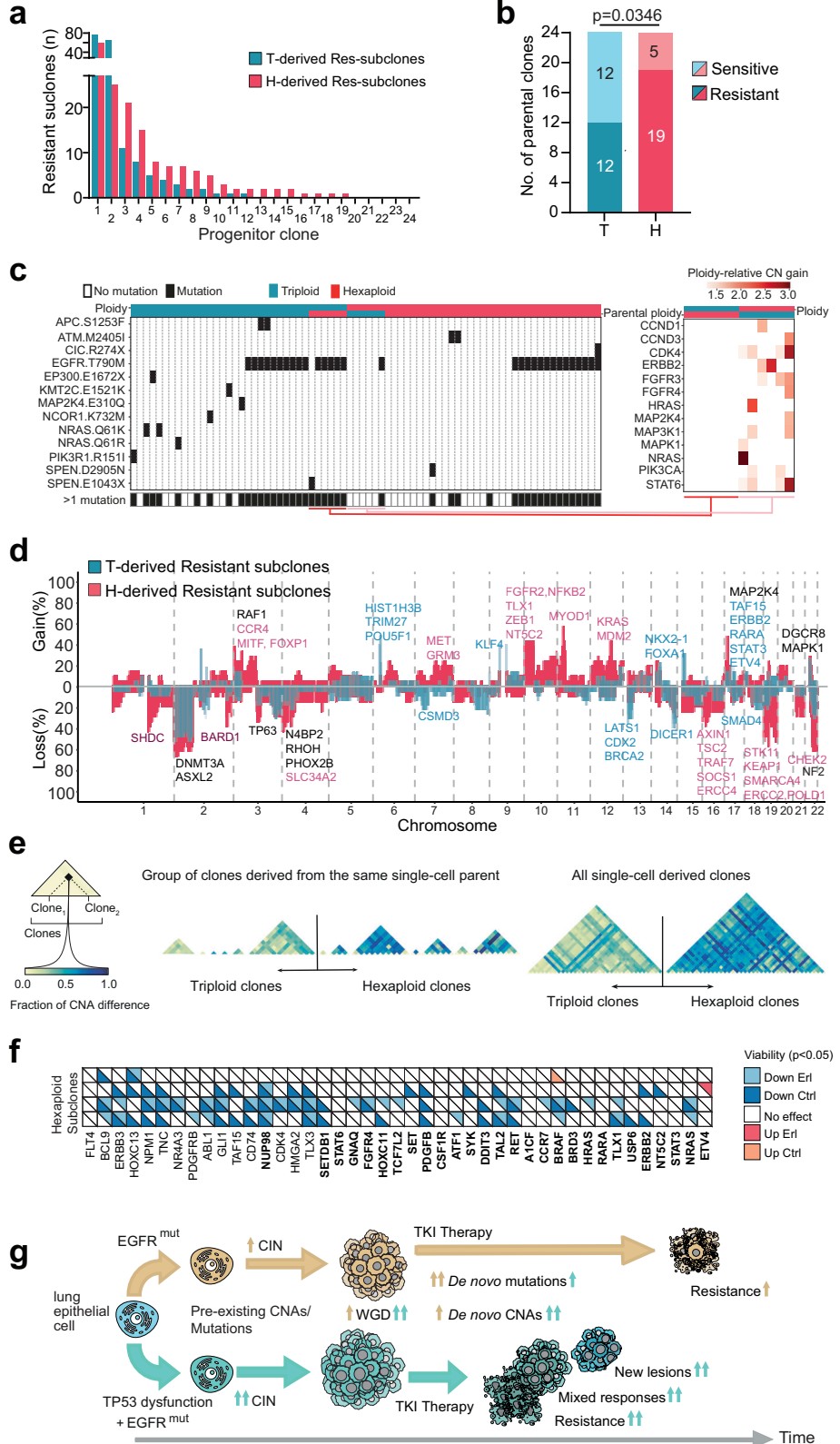

resistance compared to hexaploid, genome-doubled clones (19/40, $p = 0.021$, Chi-squared test),

Consistent with data presented for the EP mouse tumors (Supplementary Fig. 7a–c), WGD with p53 dysfunction was also found to be significantly associated with an increase in the frequency of SCNA acquisition across the genome in resistant PC9 hexaploid cells (Fig. 4d and Supplementary Fig. 9a, Paired Wilcoxon test, $p < 2.2e^{-16}$ for both

gains and losses, see Supplementary Data 3 for significantly gained or lost genes in the two genotypes). Furthermore, we found a significant difference in clone-to-clone diversity (measured as the fraction of the genome with different SCNAs between every clone pair) between hexaploid-resistant clones and triploid-resistant clones (Fig. 4e and Supplementary Fig. 9b $p = 6.611e^{-263}$, two-sided Mann–Whitney $U$-test). The results from the isogenic PC9 system reflect an increased plasticity

**Fig. 4 | Genome doubling permits elevated ploidy and promotes multiple avenues to therapy resistance in the presence of p53 pathway dysfunction. a** Plot showing the number of resistant subclones generated from each of the 24 triploid and 24 hexaploid progenitor clones after 5 weeks of culture in 1.5 µM erlotinib. **b** Number of triploid (blue) and hexaploid (red) progenitor PC9 clones that generated at least one erlotinib-resistant subclone ($p = 0.0346$, chi-squared test). **c** Left panel: Presence of somatic mutations in genes related to the EGFR pathway (black squares) is reported across all triploid (upper blue row) and hexaploid (upper red row) resistant daughter clones derived from either triploid (lower blue row) or hexaploid (lower red row) parental clones. Right panel: Ploidy-relative copy-number gains (red colors) are reported for resistant daughter clones that have changed their ploidy state for 13 genes whose gain is known to have a role in TKI resistance. **d** Frequency of copy-number gains (positive y-axis) and losses (negative y-axis) are reported across either triploid (blue) or hexaploid (red)

resistant daughter clones, highlighting events affecting oncogenes and tumor suppressors. **e** Clone-to-clone diversity measured by computing the copy-number difference (fraction of genome with different copy numbers reported in blue colors) between either left all pairs of triploid and hexaploid resistant daughter clones derived from the same parental clone (triangles), or right all pairs of triploid and hexaploid resistant daughter clones. **f** Impact of siRNA mediated repression of gained genes on re-sensitization of erlotinib-resistant hexaploid PC9 subclones. By factoring in effects on viability, the effect of gene silencing on erlotinib resistance was scored. Tiles corresponding to genes exhibiting a significant treatment-varying response upon knockdown ($p < 0.05$) in a hexaploid subclone are colored. Hue corresponds to the direction of change, and brightness to the erlotinib treatment status. Gene names depicted in bold did not impact parental PC9 viability. **g** Model depicting factors contributing to tumor resistance in E and EP tumors. Source data are provided as a Source Data file.

and ability to modify copy-number states in the resistant hexaploid clones compared to triploid clones, suggesting that WGD is an alternative mechanism to generate resistance. Consistent with the hypothesis that WGD events are positively selected for during the development of EGFR TKI resistance in p53 mutant cells, flow-cytometry analysis revealed that 22.6% (12/53) of resistant clones originating from triploid progenitors had increased their ploidy more than 1.5 times, (median of increased ploidy ≈6, range ≈4.5–7.5), whereas only 1% of the resistant hexaploid clones further increased their ploidy more than 1.5 times (one subclone out of 101 clones analysed Supplementary Fig. 9c).

Based on these findings, we would expect triploid clones, without a known somatic mutation event, to acquire ploidy gains and SCNAs as their mechanism of resistance. Consistent with this hypothesis, only one of the six triploid clones investigated that underwent a WGD event during the acquisition of EGFR TKI resistance developed a T790M mutation. The remaining 5/6 previously triploid clones had significantly higher numbers of copy-number gains in genes associated with TKI resistance, such as *FGFR3*, *NRAS*, and *ERBB2*, compared to clones which remained triploid after having acquired resistance (Fig. 4c, right panel, $p < 0.00001$, two-sided Chi-squared test). Conversely, six hexaploid clones, that displayed a reduction in their ploidy to a triploid state, harbored a point mutation as a mechanism of resistance (five subclones; T790M, one subclone; *SPEN* E1043X) following the acquisition of EGFR TKI resistance.

These data support the role of WGD on a *TP53* mutant background in the expansion of genomic opportunities for the development of treatment resistance, primarily through SCNA-based mechanisms and, less frequently, through point mutation-driven resistance mechanisms following a reduction in tumor ploidy status. Since the resistance of hexaploid PC9 subclones and EP mouse tumors is likely driven by SCNAs at a high frequency, we compared the copy-number profiles of hexaploid PC9-resistant subclones with those of EP-resistant mouse tumors through a synteny analysis to investigate the presence of shared recurrent SCNA-driven mechanisms. We identified the presence of several genes previously implicated in TKI resistance altered in both model systems, such as *KRAS*, *SRC*, and *BRAF* (Supplementary Fig. 10a, gains; b, losses).

To validate potential copy-number mechanisms of erlotinib resistance in PC9 cells, a functional siRNA screen was performed in four hexaploid-resistant subclones with distinct copy-number changes (Supplementary Fig. 10c) and in the parental PC9 cells. We used siRNA to silence 43 genes significantly gained in any one of the populations of resistant hexaploid subclones that had been sequenced. Cell numbers were scored by DAPI staining and scanning in a CellInsight CX7 High content platform after 5 days of erlotinib treatment.

Erlotinib resistance was assessed by comparing cell numbers following gene silencing in the presence or absence of erlotinib (see Methods, Fig. 4f). In total, of the 43 gained genes investigated, 10, including *NRAS*, *ERBB3*, *HRAS*, *BRAF*, *NRAS*, *HRAS* and *BRAF* led to EGFR

TKI-re-sensitization ($p < 0.05$, light blue triangles) in at least one hexaploid subclone after siRNA mediated knockdown (Fig. 4f) suggesting that, depending on the subclone, copy-number gains in these genes might contribute to TKI resistance and that different resistance mechanisms might be adopted by the individual resistant subclones in line with reported results[51].

These clinical, in vivo and in vitro data indicate that loss of p53 in the context of mutated *EGFR* and WGD, leads to an altered and malleable genomic landscape which accelerates the evolution of SCNAs under selection pressure, such as that imposed by targeted therapy. In turn, this facilitates the emergence of resistant subclones more rapidly through the acquisition of SCNAs encompassing genes functionally implicated in drug resistance and, less commonly, following a reduction in ploidy, through SNV-based mechanisms (Fig. 4g). Clinically, TP53 dysfunction with WGD likely expands the potential routes to therapy resistance during clonal evolution, contributing to earlier treatment failure, that can manifest as mixed responses to therapy within individual patients.

## Discussion

Despite the clinical efficacy of EGFR TKI therapy in oncogenic *EGFR* mutation-driven LUAD, resistance develops in the majority of patients. This is associated with additional oncogenic mutations and SCNAs[13,14,21]. Except for gatekeeper mutations such as T790M, successfully targeting these resistance mechanisms has been challenging, and in a large proportion of patients, a clear resistance mechanism is not always evident. Our analyses indicate that there is greater complexity in the response of tumor lesions than is evident from conventional RECISTv1.1 definitions. We demonstrate that mixed responses, where there are both responding and progressing lesions within an individual patient, are common in patients with NSCLC treated with either chemotherapy or EGFR TKI therapy, and are likely associated with reduced clinical benefit. Understanding the mechanisms underlying mixed responses may help identify new therapeutic approaches to forestall resistance, including both novel systemic therapies or early intervention with ablative local therapies.

CIN, WGD, and an increased prevalence of SCNAs have all been associated with a worse prognosis in several tumor histologies[22,28,52]. Our work demonstrates the plasticity rendered by p53 dysfunction together with WGD in driving a diversity of resistance mechanisms through somatic mutations and SCNAs. We observed a high degree of concordance between the human and murine datasets suggesting that the GEMMs used in this study provide relevant models to study clonal evolution in human *EGFR*-driven LUAD in both the naïve and EGFR TKI-treated settings. We identify syntenic genomic regions affected by SCNAs in both mice and patients that may contribute to the early development of resistance observed following TKI therapy in EP tumors. For example, *PTEN* was significantly lost in naïve EP, but not E tumors (Fig. 2a), and *PTEN* loss has been shown to contribute to TKI resistance[36].

A recent study revealed that WGD occurs in almost 30% of all sequenced tumors[22] and we have recently proposed WGD as a mechanism that mitigates the accumulation of deleterious mutations[23]. Our in vivo models indicate that although E tumors frequently exhibited WGD, these tumors predominantly had homogenous responses to treatment. These data suggest that WGD alone does not promote mixed responses in tumors with a functioning p53 signaling pathway (Figs.1e, 2c). Instead, the isogenic model data presented here demonstrates that it is in the context of p53 dysfunction that WGD broadens the potential routes to acquired resistance by increasing somatic copy-number diversity. Importantly, an acute genome doubling event in PC9 cells on the background of p53 dysfunction is insufficient for EGFR TKI resistance. Resistance only emerges more frequently after 5 weeks of TKI exposure in these cells, a similar time course to the first CT scan after initiating TKI therapy in the AURA trials.

These data suggest that in addition to common *EGFR* resistance mechanisms such as T790M, the combination of WGD and p53 dysfunction provides the SCNA diversity required to generate resistant clones capable of withstanding the selection pressure generated by EGFR TKI therapy. We identified that the knockdown of several genes gained in resistant PC9 hexaploid subclones conferred erlotinib re-sensitization, many of which have been previously associated with *EGFR/RAS/PI3K* pathway activation in different systems[13,53]. Taken together, our data suggest a vital role for *TP53* loss in permitting subsequent mixed somatic copy-number evolution following a WGD event, thereby expanding the possible routes to erlotinib resistance, resulting in early treatment failure and contributing to the dynamic nature of lesion-to-lesion response within individuals. Our data confirms the association of reduced overall survival in the context of *EGFR* and *TP53* co-mutations and suggests that the presence of TP53 pathway alterations in *EGFR*-driven lung cancer might act as a surrogate marker of CIN and identify patients at increased risk of mixed tumor responses to TKI therapy and earlier progression.

Our clinical analyses of a total of 508 lesions from 99 patients is not without limitations: the tissue samples analysed were taken from three different trials (AURA trial phase 2 expansion cohort, AURA2 trial, and AURA3 trial) and were limited to those patients with tissue-based somatic tumor analyses as well as available imaging for longitudinal analyses. However, there was no difference in clinical characteristics between patients with E and EP tumors (Supplementary Fig. 2b). Furthermore, the PFS times observed with this smaller cohort are comparable to those seen in the AURA3 study. Finally, although data based on plasma analysis of *TP53* from the AURA3 trial showed minimal differences in PFS between *TP53* mutant and wild-type patients[35] contrary to our tissue based analysis, clonal hematopoiesis of indeterminate potential (CHIP) complicates plasma mutation calls (particularly for *TP53*) and calling *TP53* copy number loss from ctDNA is challenging.

In conclusion, these findings demonstrate that EGFR activation, together with TP53 pathway inactivation and WGD, remodels the copy-number landscape to create an environment which is permissive for the development of diverse mechanisms of resistance to TKI therapy resulting in mixed response dynamics in vivo. A better understanding of SCNA-driven resistance mechanisms is required to develop strategies that improve outcomes in this setting where high levels of CIN are tolerated. Successful approaches may involve a combination of local therapy to remove or ablate sources of complex genotypes that contain TKI-resistant clones together with a combination of drug treatments to target SCNA-driven resistance. While our study focused on *TP53* mutations in *EGFR*-mutated LUAD, we propose that the conclusions drawn with respect to response heterogeneity may be generally applicable in tumors with a clonal actionable driver oncogene and loss of p53 function. Our findings suggest that assessing TP53 status may guide more informed discussions regarding TKI success rates, and the potential clinical benefit of frequent disease monitoring.

## Methods

All regulated animal procedures were approved by The Francis Crick Institute BRF Strategic Oversight Committee, incorporating the Animal Welfare and Ethical Review Body, conforming with UK Home Office guidelines and regulations under the Animals (Scientific Procedures) Act 1986 including Amendment Regulations 2012. The TRACERx observational study (NCT01888601) has an approval from the UK research and ethics committee (13/LO/1546).

### Animal procedures
Animals were housed in ventilated cages with unlimited access to food (2018 Autoclavable Rodent Breeding Diet, ENVIGO RMS UK LTD, T.2018S.12) and water. *EGFR*-L858R [Tg(tet-O-EGFR∗L858R)56Hev][54] mice were obtained from the National Cancer Institute Mouse Repository. R26tTA [Gt(ROSA)26Sortm1(tTA)Roos][55] and *Trp*53fl/fl [Trp53tm1Brn][56] mice were obtained from Jackson laboratory. Mice were backcrossed onto a C57Bl6/J background and further crossed to generate Rosa26tTaLSL/tet(O)*EGFR*L858R and Rosa26tTaLSL/tet(O)*EGFR*L858R/*Trp53*flox/flox mice. After weaning, the mice were genotyped (Transnetyx, Memphis, USA) and placed in groups of one to five animals in individually ventilated cages, with a 12-h daylight cycle. Recombination (animal age 2–6.5 months) was initiated by adenoviral CMV-Cre (Viral Vector Core, University of Iowa, USA) delivered via intratracheal intubation (single dose, $2.5 \times 10^7$ virus particles/50 µl). The animal cohorts used for experiments were balanced for sex.

### Micro-CT imaging
For tumor emergence, tracking and measurements, the thorax was scanned once a month using a Bruker, Skyscan 1176. Mice were anesthetized using isoflurane, and the acquired CT images were processed using RespGate for respiratory gating and NRecon for z-stack image reconstruction. For tumor diameter measurement, volume calculation and viewing, we used a combination of CT-Analyser and DataViewer. The final resolution of reconstructed z-stacked images was 50 µm/pixel. An object was deemed a tumor if its measured diameter was at least 300 µm and if, by comparing two consecutive monthly CT scans, the size had increased or was absent in the previous scan. No regulated procedures undertaken in this project exceeded the permitted limits (10% weight loss over a 24 h period). Mice were sacrificed immediately after observing any difficulty breathing or if projected/expected to start showing distress before the next CT scan. Observation of difficult breathing included when a mouse was under general anesthesia while micro-CT scanning. Including general anesthesia in abnormal breathing distress also enabled us to maintain high and consistent quality of micro-CT images and tumor volume and diameter measurements. Calculated diameters were analysed using GraphPad Prism 7 for statistical graph design.

### Tissue harvest, histology, tumor burden analysis, image cytometry
After sacrifice, lung tissue was immediately removed, and individual tumor nodules were isolated from one lung lobe. In order to generate single cells, tumors were cut into small pieces and incubated in 1 ml collagenase/dispase (MERCK, 102696380011, 1 mg/ml in PBS) at 37 °C for 15 min with continuous shaking. Tumor material was pipetted up and down until able to pass easily through a p1000 tip, allowed to sediment, and the supernatant was removed and placed on ice. A fresh aliquot of collagenase/dispase was added to the tissue, and the samples were incubated for an additional 15 min. The material was combined and passed through a 100 µm cell strainer before washing the cells once in PBS and freezing in 90%FBS/10%DMSO. The remaining lung lobes were fixed overnight in 10% neutral buffered formalin, transferred to 70% ethanol and processed for paraffin embedding. Tissue sections (4 µm) were stained with H&E or immunostained with anti-Ki67 (Abcam, ab15580) using the ROCHE Ventana platform,

antibody dilution 1:1000 with 24 min CC1 antigen retrieval and OM anti-rabbit HRP (05269679001). Lung and tumor area quantifications were carried out on H&E-stained slides. Tumor grading was carried out by a trained pathologist according to published criteria[57]. Paraffin-embedded blocks were used for image cytometry analysis. Formalin-fixed paraffine-embedded tumors were used for the preparation of nuclei suspensions, the nuclei were stained with Feulgen−Schiff. The samples were analyzed with the Fairfield DNA Ploidy system (Fairfield Imaging, Kent, UK) which measures the optical density and nuclear area. The integrated optical density of each nucleus was calculated, with lymphocytes used as internal reference cells to determine the position of the diploid fraction[58].

## Mouse therapy regimens

Mice were weighed weekly and treated with erlotinib (ERL; 5 mg/ml in 0.3% methylcellulose/H2O, Mon-Fri, 25 mg/kg) or chemotherapy (mixed suspension of carboplatin and paclitaxel, 3.33 and 0.66 mg/ml respectively) via intraperitoneal injection (See Supplementary Table 4 for mouse treatments). Mice began therapy upon identifying at least one lung tumor with a minimal diameter of 1 mm. If multiple smaller tumors were found (granular appearance of lungs), mice were scanned again after 2 weeks, and if it was thought that the animal's welfare could be compromised within the next 2 weeks, therapy was initiated. Mice initially treated with chemotherapy (4 weeks, two doses per week, 16.65 mg/kg carboplatin and 3.3 mg/kg paclitaxel) were allowed to recover for one month before starting ERL treatment. In cases where CT scans showed small or no treatment response, ERL was given immediately. For generating resistant tumors, an alternating monthly ERL on and off schedule was changed to continuous therapy after the detection of resistance by micro-CT.

## DNA purification and processing

Genomic DNA was purified using AllPrep DNA/RNA Mini Kit (Qiagen) from cells, fresh frozen tissue, and matched-normal control tissue (tail), following the manufacturer's recommendations. After an initial quality control by gel electrophoresis, DNA was quantified using Qubit™ dsDNA BR Assay Kit (Thermo Fisher Scientific) and BioAnalyzer.

## Whole-exome sequencing−mouse data

WES was performed by the Advanced Sequencing Facility at The Francis Crick Institute using the Agilent SureSelectXT Mouse All Exon Kit for library preparation. Sequencing was performed on HiSeq 4000 platforms.

**Alignment—mouse.** All samples were de-multiplexed and the resultant FASTQ files aligned to mm10 using bwa-mem (bwa v0.7.15). De-duplication was performed using Picard (v2.1.1) (http://broadinstitute.github.io/picard). Quality control metrics were collated using FASTQC (v0.10.1−http://www.bioinformatics.babraham.ac.uk/projects/fastqc/), Picard, and GATK (v3.6). SAMtools (v1.3.1) was used to generate mpileup files from the resultant BAM files. Thresholds for base phred score and mapping quality were set at 20. A threshold of 50 was set for the coefficient of downgrading mapping quality, with the argument for base alignment quality calculation being deactivated. The median depth of coverage for all samples was 92x (range: 58–169x).

**Variant detection and annotation—mouse.** Variant calling was performed using VarScan2(v2.4.1), MuTect(v1.1.7), and Scalpel(v0.5.3)[59–61].

The following argument settings were used for variant detection using VarScan2:

--min-coverage 8 --min-coverage-normal 10 --min-coverage-tumor 6 --min-var-freq 0.01 --min-freq-for-hom 0.75 --normal-purity 1 --p-value 0.99 --somatic-p-value 0.05 --tumor-purity 0.5 --strand-filter 0.

For MuTect, only "PASS" variants were used for further analyses. With the exception of allowing variants to be detected down to a VAF of 0.001, default settings were used for Scalpel insertion/deletion detection.

To minimize false positives, additional filtering was performed. For single-nucleotide variants (SNVs) or dinucleotides detected by VarScan2, a minimum tumor sequencing depth of 30, variant allele frequency (VAF) of 5%, variant read count of 5, and a somatic $p$ value <0.01 were required to pass a variant. For variants detected by VarScan2 between 2 and 5% VAF, the mutation also needs to be detected by MuTect.

As for insertions/deletions (INDELs), variants need to be passed by both Scalpel ("PASS") and VarScan2 (somatic $p$ value <0.001). A minimum depth of 50x, 10 alt reads, and VAF of 2% was required.

For all SNVs, INDELs and dinucleotides, any variant also detected in the paired germline sample with more than five alternative reads or a VAF greater than 1% was filtered out.

The detected variants were annotated using Annovar[62].

## Human *EGFR* transgene amplicon sequencing of mouse tumors

FASTQ files were aligned to hg19 obtained from the GATK bundle (v2.8) using bwa-mem (bwa v0.7.15)[63,64]. Analyses were performed using R (version 3.3.1) and the bam2R function of the deepSNV (v1.18.1) R library[65]. The median depth of coverage of sequenced EGFR exons (19,20,21) was 5290x (range: 2238-8040). Variants associated with resistance to EGFR tyrosine kinase inhibitors were queried using deepSNV's bam2R function, with the arguments q = 20 and s = 2. The variants explored include: T790M, D761Y, L861Q, G796X, G797X, L792X, and L747S. L858R, the driver mutation in the mouse model used, was identified in every sequenced sample.

## Synteny analysis

To perform synteny mapping, we leveraged the genomic ranges R package with homology mapping from the human to the mouse genome (Synteny Portal[66]) to calculate the rate of gains and losses observed in mouse tumors in each homogenous region of the human genome.

## Identification of recurrent SCNAs

The sampling and simulation method proposed by ref. 25 was used to identify recurrent SCNAs in different cohorts with inferred copy-number profiles in this study. Briefly, given a cohort of $N$ tumors with inferred copy-number profiles, the rate of gains $Rg,t$ and the rate of losses $Rl,t$ in each tumor $t$ is estimated as the fraction of the genome affected by related events. Using the estimated rates, the background distribution of gains and losses is obtained by performing 1000 simulations. Specifically, for each simulation, the copy-number state of each tumor $t$ is determined using a Bernoulli model with a probability of $Rg,t$ for gains and $Rl,t$ for losses, and the resulting total number of expected gains or losses is obtained by summing over all tumors. Using the simulated total numbers of gains and losses across all simulations, background empirical distributions are computed as well as a 95% confidence interval, which is used to define a threshold on frequencies for defining recurrent gains and losses, respectively. To determine the deviation from euploidy as a measure for SCNA amplitude, we calculated the binwise non-absolute deviation from copy number state 2 across all curated single-cell WGS libraries. The mean deviation was then plotted across the genomic positions by genotype (E vs EP) and by treatment group (naïve vs resistant).

## Single-cell whole-genome sequencing

Lung tumors were either snap frozen as whole tissue, or homogenized into single-cell suspensions and frozen in FBS with 10% DMSO. Samples were stored at −80 °C until further processing. For single-cell analysis,

we performed single-cell DNA sequencing using an established protocol[67,68].

**Preparation.** To isolate nuclei for flow sorting from frozen tissues, samples were dissociated by pushing small tissue fragments through a 70 μm strainer using a syringe plunger in nuclei isolation buffer (10 mM Tris-HCl pH 8.0, 0.32 M sucrose, 5 mM CaCl$_2$, 3 mM Mg(Ac)2, 0.1 mM EDTA, 1 mM DTT, 0.1% Triton X-100 (v/v)). Nuclei were spun down at $1000 \times g$ for 5 min at 4 °C and resuspended into PBS with 2% BSA, 10 μg/mL Hoechst 33258, and 10 μg/mL propidium iodide (PI). Single-cell suspensions were prepared for flow sort by resuspension in nuclei staining buffer (100 mM Tris-HCl pH 7.5, 154 nM NaCl, 1 mM CaCl2, 0.5 mM MgCl$_2$, 0.2% BSA, 0.1% NP40 (v/v), 10 μg/mL Hoechst 33258, and 10 μg/mL PI). Isolated nuclei suspensions were collected into FACS tubes with 70 μm strainer caps. Both frozen tissue and single-cell suspension samples were incubated on ice for at least 15 min prior to flow sorting. Intact single nuclei from predetermined DNA populations were sorted using a FACSJazz (BD Biosciences) into 96-well plates containing ProFreeze-CDM (Lonza) buffer and 7.5% DMSO. Plates were sealed and centrifuged at $500 \times g$ for 5 min at 4 °C and stored at −80 °C until library preparation.

For library preparation and sequencing, DNA was fragmented using micrococcal nuclease (MNase) followed by end-repair, A-tailing, and Illumina adapter ligation. AMPpure XP beads and 80% ethanol were used for clean-up steps between reactions. Barcoding and library amplification were performed using a multiplexing primer mix and PCR for 17 cycles. All liquid handling prior to pooling was done using a Bravo Automated Liquid Handling Platform (Agilent). Libraries were subsequently pooled, cleaned using ethanol, and size-selected using 2% E-Gel EX agarose gels with SYBR Safe Stain. Mono- and dinucleosomal bands were excised, and DNA was isolated using a Zymoclean™ Gel DNA Recovery kit. Libraries were quantified using a Qubit fluorometer (Thermo Fisher Scientific), and library fragment size distributions were assessed on a Bioanalyzer (Agilent). Library pools were diluted to 2 nM and sequenced on an Illumina NextSeq500 at ERIBA (Groningen, The Netherlands). FASTQ-files were generated using standard Illumina software (bcl2fastq v1.8.4).

**Analysis.** De-multiplexed FASTQs were aligned to the mouse genome mm10 using Bowtie2 (v2.2.4)[69]. Duplicate reads were removed using BamUtil (v1.0.3)[70]. Single-cell copy-number profiles were generated using AneuFinder (v1.8.0)[70]. The ploidies of the cells were estimated using flow-cytometry analyses described above. AneuFinder was permitted to find copy-number solutions with a ploidy range of ±0.5 of the flow-cytometry estimated ploidy (the "most.frequent.state"). The "edivisive" method was used for copy-number detection. Blacklisted regions were generated from the published euploid reference[67]. Copy-number profiles were generated with bin sizes of 2 Mb and GC correction. The number of random permutations was set at 20; and the sig.lvl argument was set at 0.05.

For quality control of scDNA-seq, we used several metrics automatically generated by the AneuFinder, as well as additional metrics. Additional QC metrics included the median absolute deviation of coverage in each copy-number (CN) segment and the median deviation from the coverage equivalent to the copy number called in each CN segment. First multi-variant clustering is performed (clusterByQuality function from the AneuFinder package) on the automatically calculated metrics by AneuFinder using all cells in the study. We excluded any cell clusters in which the average read count in each 2 megabase (Mb) bin was <100 and the SOS (sum of squares between raw and scaled CN profiles) was $>3 \times 10^6$. We then performed additional filters on each of the remaining cells using read count, SOS, entropy, spikiness, complexity, and our own QC metrics (see associated code). Finally, we performed a visual QC of the remaining cells and removed an additional 63 cells where the raw sequencing coverage poorly fit the calculated copy number. This was caused by either highly variable sequencing coverage within bins or due to incorrect ploidy estimation in these cases. In total, 499/2448 cells (18%) were removed during the quality control process.

**Filtering and QC.** Using the inferred single-cell copy numbers, we filtered and excluded cells from two specific groups of cells. Firstly, we identified and excluded from downstream analysis normal diploid cells (likely corresponding to normal epithelia contamination or infiltrating lymphocytes) as any cell with no or very few SCNAs, that is, a fraction of the genome <5% with a total copy number of 2. The same filtering threshold has been applied to cells with <5% of total copy numbers of 4 as these cells might correspond to in G2 cell cycle phase. Second, cells with noisy inferred copy-number profiles are frequent in single-cell sequencing due to the presence of cells in the S-phase of the cell cycle with actively replicating DNA (12–42%), cells with a low number of sequenced reads (-8%), and doublets (>2%). To prevent an impact on downstream analysis, we have identified cells with noisy profiles as any cell that only shared <33% of SCNAs breakpoints or whole-chromosomal aberrations with the other cancer cells from the same tumor. After excluding normal diploid cells and cancer cells with noisy copy-number profiles, the remaining cells were used for downstream analysis. On this basis, after sequencing, data from 2/9 E-resistant mice were excluded from further analysis due to the low quality of resulting cells.

## Human clinical survival and imaging data

For all imaging analysis, the criteria for a mixed/heterogeneous response was defined by Reiter and colleagues[15]. The RECIST database was queried for NSCLC patients who had at least two lesions, one of which shrank by 30% or more, and these patient's RECISTv1.1 measurements were subsequently used for the analysis of response to erlotinib and cytotoxic chemotherapy. Informed consent for all patients within the AURA studies used for this analysis was taken by the sponsor (AstraZeneca) for the clinical trial activity and for the sharing of sequencing and imaging data with external collaborators. Patients from the AURA cohorts (Suppl. Table 3) were combined for the relevant analysis. For the survival analysis all patients had somatic *TP53* and *EGFR* mutation status assessed by the FoundationOne commercial assay from Foundation Medicine. Only pathogenic mutations were used to assign the relevant genotype. Patients who had consented to share imaging and had at least two measurable lesions were included in the subsequent analysis (see consort diagram). DIACOM files containing CT axial imaging performed with contrast were reviewed for each patient. All measurable lesions were included irrespective of whether they were defined as target lesions. The measurements of lesion dimensions were performed by two clinical oncologists (MS and CH). RadiAnt DICOM Viewer 5.5.0 software was used. The longest diameter of each lesion from axial imaging was summed and compared to the baseline osimertinib CT scan. The percentage change from baseline was assessed at the first scan (-12 weeks since commencement of osimertinib) and at the time of maximum response. We adopted the Reiter et al thresholds for grading the degree of response of an individual lesion to define it as responding, stable, or progressing. Two time points were used for determining mixed responses in this study: (i) the response of all lesions at the first follow-up scan performed 6–8 weeks after the beginning of the treatment or (ii) the best response ever achieved by a lesion was used to call the heterogeneity of response. Homogeneous response was recorded if at least one lesion attained at least partial response, i.e., ≥30% shrinkage of the largest diameters, and the remaining lesions did not increase in size by more than 10%, and there were no new lesions. Otherwise, a mixed response was recorded.

The EORTC RECIST database was queried for mixed responses as outlined in the text. The response assessment closed to 12 weeks following initiation of treated was used for the analysis.

## Additional AURA2/3 patient treatment information

Patients in the AURA3 trial were previously treated only with a first generation of EGFR TKI (gefitinib or erlotinib), whereas patients in the AURA2 trial and the AURA-extension cohorts additionally received one or more lines of chemotherapy before switching to osimertinib. The subset used for the overall survival analysis included patients who were selected because of a poor response to osimertinib (defined as those with progressive or stable disease only or those with partial or complete response but a PFS of less than six months). For this subset, pretreatment targeted sequencing data were analysed; patients with deleterious mutations in *TP53* (affecting splice sites, DNA binding, transactivation domains, and tetramer binding), *TP53* deletion, or MDM2/4 amplification were defined as having p53 pathway disruption (EP; 82 of 117).

## University of California, San Francisco (UCSF) clinical cohort analysis

We analyzed patients with metastatic non-small cell lung cancer (NSCLC) with EGFR exon 19 deletions, L858R, or T790M mutations who received osimertinib therapy between 2015 and 2021. Tumor genomic analysis was conducted using the UCSF 500 Cancer Gene test, which employs next-generation sequencing to detect somatic alterations in a panel of 529 cancer genes.

Patients were stratified based on tumor genomics: (1) those with TP53 pathway disruption, including p53 mutations and MDM2 amplifications. (2) those with wild-type TP53. We analyzed treatment response by assessing radiographic changes in the size of individual malignant lesions. We compared the largest diameter of each lesion between pretreatment imaging and the first surveillance scan while also noting the emergence of any new lesions.

## Cell culture

PC9 cells were obtained from Cell Services at The Francis Crick Institute, UK, where short tandem repeat profiling and mycoplasma testing is routinely performed to ensure cell identity and quality. The STR testing for the batch of banked PC9 cells used in this project was performed on 22/07/2020 and 12/08/2020. Cells were maintained at 37 °C in 5% $CO_2$ in RPMI or Dulbecco's Modified Eagle Medium (DMEM) with high glucose and L-glutamine, respectively (Invitrogen), supplemented with 10% FBS, 1× PenStrep (Sigma). Cells were incubated with 10 μg/mL Hoescht 33342 (Sigma) for 1 h at 37 °C. Cells were single-cell sorted and assessed for ploidy (Supplementary Fig. 8a).

## Flow-cytometry

Clonal cell populations were cultured in 10 cm dishes until ~60% confluency, harvested by trypsinization (0.05% trypsin, Thermo Fisher Scientific), washed with PBS, and fixed/permeabilized by drop-wise addition of 2 ml 70% ethanol while stirring and stored at 4 °C until further use. On the day of analysis, fixed cells were washed twice with PBS and stained using 50 mg/ml propidium iodide solution (PI; up to 2 ml per cell pellet), filtered through 30 mm nylon mesh into 5 ml round bottom polystyrene tubes (Corning) and incubated at 4 °C overnight. DNA index of cell populations was measured the following day using a BD LSR Fortessa flow cytometer. Fixed parental PC9 cells were used as controls for all cytometry analysis batches. The same gating strategy (Supplementary Fig. 8a) was used in all experiments, and the analysis and inference of ploidy was performed using FlowJo10 software.

## Erlotinib dose-response curve

Cells were seeded, 2000 cells per well, in 100 μL, in triplicate in 96-well, black, transparent flat-bottom plates, and treated with erlotinib (dose 700–0 μM) for 96 hours in standard culturing conditions. Cell viability was assessed using CellTiter-Blue(Promega) according to the manufacturer's recommendations. Fluorescence was measured using an EnVision 2102 MultiLabel Reader, at 560Ex/590Em nm. All measurements were normalized against background fluorescence. Final data analysis, graphing, calculation of IC50, and statistical analysis was performed using Microsoft Excel and GraphPad Prism software. Biological replicates were separated by one cell passage and data from three biological replicates were combined for the calculation of IC50 and plotting of the dose-response response curve per cellular group.

## Generating erlotinib-resistant subclones

We randomly selected 48 progenitor clones (24 triploid and 24 hexaploid) and plated each clone into one 96-well, flat-bottom culture plate, at a density of 5000 cells per well. Cells were cultured in standard conditions in 200 μl complete media per well, supplemented with 1.5 μM erlotinib. The media was changed twice per week. After 5 weeks of incubation, plates were inspected under the microscope, and wells with viable colonies were labeled for expansion. Expansion followed a standard protocol of passaging through 24-well and 6-well plates to 10- and 15-cm culture dishes (Falcon). All subclonal populations were expanded with media containing 1.5 μM erlotinib.

## Whole-exome sequencing—PC9 cell lines

WES was performed by the Advanced Sequencing Facility at The Francis Crick Institute using the Twist BioScience Agilent Human Core Exome Kit for library preparation. Sequencing was performed on HiSeq 4000 platforms.

**Alignment—PC9 cell line.** All samples were de-multiplexed and the resultant FASTQ files aligned to the hg19 genome using bwa-mem (bwa v0.7.15). De-duplication was performed using Picard (v1.107) (http://broadinstitute.github.io/picard). Quality control metrics were collated using FASTQC (v0.11.5- http://www.bioinformatics.babraham.ac.uk/projects/fastqc/), Picard, and GATK (v3.6). SAMtools (v1.3.1) was used to generate mpileup files from the resultant BAM files. Thresholds for base phred score and mapping quality were set at 20. A threshold of 50 was set for the coefficient of downgrading mapping quality, with the argument for base alignment quality calculation being deactivated. The median depth of coverage for all samples was 155x (90-251x).

**Variant detection and annotation—PC9 cell lines.** Variant detection was performed using MuTect2 (GATK v4.1.3) using a tumor-only variant calling workflow (https://docs.gdc.cancer.gov/Data/Bioinformatics_Pipelines/DNA_Seq_Variant_Calling_Pipeline/).

First, OXOG (oxidation of guanine to 8-oxoguanine) artifact metrics were calculated using the *CollectSequencingArtifactMetrics* command. Pileup summaries for all the cell lines were created using the *GetPileupSummaries* command, with the gnomad common biallelic SNPs provided (https://gnomad.broadinstitute.org/downloads). Contamination metrics were calculated using the *CalculateContamination* command. Variant calling was performed using MuTect2, with the gnomad germline reference provided, as well as a panel of normal samples created by 4136 TCGA curated normal samples (gatk4_mutect2_4136_pon.vcf.gz). The resulting VCF was sorted using Picard (v2.18.11). The *FilterMutectCalls* command was used to filter any contaminated variant calls identified from the *CalculateContamination* step. Additional orientation bias filtering was performed using the *FilterByOrientationBias* command. Variants that failed MuTect2 filtering were excluded from downstream analyses (variants identified as "clustered_events", "slippage", "weak_evidence", "base_qual", "strand_bias", "contamination"," multiallelic", "map_qual", "position", "fragment"). To further minimize false calls, a minimum variant depth of coverage of 50x was needed, with more than 10 reads supporting the alternate allele for INDELs/multi-nucleotide variants, 5 reads supporting the alternate allele for SNVs, and a variant allele frequency of more than 2% being required. Variant annotation was performed using Annovar. Variants with an SNP frequency in the population of more

than 5% were filtered from downstream analyses (using exac database and 1000genomes project). Driver mutations in cancer genes (genes found in the cancer gene census (v90)[70,71] and a pan-cancer consensus oncogene list generated by ref. [72]) were explored as possible mutation mechanisms of resistance.

**Somatic copy number aberration detection—PC9 cell lines.** Bed-Tools (v2.26) was used to extract the read counts across 1 Mb bins in the sequenced progenitor and resistant cell lines using the *multicov* function. Bins with no coverage in the progenitor or resistant cell lines were removed from downstream analyses.

Single SNPs were identified from the progenitor cell lines using platypus (v0.8.1). Default parameters were used apart from the *gen-Indels* flag set to FALSE. The resistant cell lines were genotyped based on the variants identified in the progenitor cell line. Only autosomal chromosome SNPs in resistant cell lines with a coverage depth of at least 40 and variant count of at least 10 were kept for downstream analyses. A two-sided binomial test was used to define SNPs as heterozygous (comparing BAF to 0.5, with $p$ value threshold of 0.05).

**Copy-number calling.** Existing copy-number calling methods require DNA sequencing of matched-normal samples for each tumor sample to be analysed, which were not available for the considered PC9 cell lines. However, for every derived PC9 subclone analysed in this study, DNA sequencing data from each corresponding progenitor PC9 clone is available. Moreover, in this study, we were only interested in identifying SCNAs that have been specifically acquired in the derived PC9 cell lines while excluding those SCNAs in the corresponding progenitor. In fact, only the former can correspond to potential mechanisms of resistance. To reach these goals, we proposed a copy-number calling approach which extends existing methods by using progenitor sequencing data instead of matched-normal sequencing data and only identifies novel SCNAs in the derived cell line. As such, the proposed method is composed of four steps.

First, DNA sequencing reads are counted in fixed-size genomic bins. In particular, BedTools (v2.26) was used to extract the read counts across 1 Mb bins in the sequenced progenitor and resistant cell lines using the *multicov* function. Bins with no coverage in the progenitor or resistant cell lines were excluded from downstream analyses.

Second, germline single-nucleotide polymorphisms (SNPs) were identified from the progenitor cell lines using platypus (v0.8.1). Default parameters were used apart from the *genIndels* flag set to FALSE. The resistant cell lines were genotyped based on the variants identified in the progenitor cell line. Only autosomal chromosome SNPs in resistant cell lines with a coverage depth of at least 40 and variant count of at least 10 were kept for downstream analyses. A two-sided binomial test was used to define SNPs as heterozygous (comparing BAF to 0.5, with $p$ value threshold of 0.05).

Third, the existing segmentation algorithm DNAcopy (v1.54) was used with default settings to identify genomic segments resulting from SCNAs. Since read-depth ratios are required as an input to this algorithm, we calculated read-depth ratios for every genomic bin as the ratio of the corresponding read counts in the resistant cell line to those in the progenitor cell line. To account for a different total number of sequencing reads that are sequenced in different samples, the read-depth ratio was further divided by the ratio of the total number of reads in the progenitor cell line to the total number of reads in the resistant cell line. As such, DNAcopy combines neighboring genomic bins that are affected by the same SCNA and are part of the same genomic segment. Moreover, an estimated read-depth ratio is provided for each genomic segment, which is a value proportional to the corresponding copy-number variation from the progenitor to the derived cell line.

Lastly, we identied genomic regions affected by copy-number gains or losses using allelic-balanced genomic regions as a reference. In fact, similar to previous copy-number studies, we assume that allelic-

balanced genomic regions (i.e., genomic regions in which every cell has the same number of copies, such as (1, 1), (2, 2), etc.) are always present. Note that this is a reasonable assumption since SCNAs do not generally affect every genomic region, and the remaining genomic regions are affected by zero or more WGDs. As such, we identify allelic-balanced genomic regions as genomic bins in which the hypothesis of allelic balance (allele frequency equal to 0.5) cannot be excluded for >20% of the putative SNPs in such bin (a threshold of 20% has been chosen to account for the presence of somatic variants and homozygous SNPs with sequencing errors). Specifically, we perform this test using a standard Binomial model for sequencing data. We thus use the read-depth ratios for all the genomic bins in allelic-balanced segments to empirically estimate the distribution of read-depth ratios for these regions. Since multiple copy-number states can underly allelic-balanced genomic region, we use a Gaussian model for read-depth ratios as in previous studies, and we separate the distributions of different copy-number states using a Gaussian use a Gaussian mixture model (edivisive method of mclust algorithm, v5.4.5). We then select the largest distribution as a reference, and we use the reference Gaussian distribution to identify lost bins using a two-sided $Z$-test. Given the expected number of false positives based on the chosen size of the fixed-size genomic bins, a significance level of 0.1% has been used. As such, genomic bins affected by SCNAs are identified as bins for which the reference distribution can be rejected, and the gained or lost status is defined according to whether the values are higher or lower than expected, respectively. Lastly, gained and lost cancer genes are selected as those found in the cancer gene census (v90)[71], and a pan-cancer consensus oncogene list is generated by ref. [72].

## Resistant PC9 subclone erlotinib re-sensitization screen

A selection of hexaploid erlotinib-resistant subclones with recurrent copy-number gains identified through WES analysis were screened to identify genes required for maintenance of resistance. Using Dharmafect 2 (Horizon Discovery), cells were reverse transfected in 96-well plates with 37.5 nM siRNA pools in the presence or absence of 1.5 μg/ml erlotinib and cell growth was monitored in an incucyte or cytomat incubator for up to 5 days until confluent. Cells were subsequently fixed with 4% paraformaldehyde and stained with 1 μg/ml dapi. Plates were scanned in a CellInsight CX7 High content platform and valid object counts were measured. Plate positional normalization using the outer product of row and column medians across the entire screen was performed to reduce the influence of edge effects. The screen was performed in triplicate and UBB was used as a positive control to assess loss of viability. Non-targeting controls were used for parental PC9 cells and resistant subclones to establish baseline conditions. For each gene, a linear model on log-transformed data was fitted in R[73] to account for a three-way interaction of perturbation status (against control), subclone and erlotinib treatment, along with a plate effect. Estimated marginal means from that model, and their standard errors, for the erlotinib treatment: knockdown interaction per subclone were used to $p$ values (unadjusted $p$-values < 0.05 were deemed statistically significant).

## Statistics and reproducibility

For analysis of patient data, no a priori sample size calculations were performed, and cohort sizes were dependent on patient data availability from both observational and clinical trials. Analysis of murine and human imaging data were performed in blinded fashion with respect to *Trp53/TP53* status respectively. For statistical analysis of cellular diversity, single-cell data from two mice was excluded due to too low coverage to reliably perform the analysis.

## Reporting summary

Further information on research design is available in the Nature Portfolio Reporting Summary linked to this article.

## Data availability

Source data are provided with this manuscript[74]. The sequencing data generated in this study have been deposited in ENA with the accession numbers PRJEB55482, PRJEB55481, and PRJEB55479 and is publicly available. Processed data (including copy-number profiles and related analysis) for the E/EP mouse tumors and for the PC9 resistance cell lines are available in Zenodo at [https://doi.org/10.5281/zenodo.10156620]. The whole-exome data (from the TRACERx study) used during this study has been deposited at the European Genome–phenome Archive, accession code EGAS00001006494. Access is controlled by the TRACERx data access committee. Details on how to apply for access are available on the linked page. Data from the TCGA and OncoSG can be found at https://www.cbioportal.org/ and https://src.gisapps.org/OncoSG/ respectively. Data from the AURA trials is available on request from AstraZeneca https://vivli.org/ourmember/astrazeneca/. Clinical parameters from the San Francisco Clinical Cohort are available upon request in a de-identified manner from Dr. Bivona or Dr. Blakely. Biological materials are available on request Source data are provided with this paper.

## Code availability

The code to reproduce the single-cell and PC9 analysis and figures in this study is available in GitHub at [https://github.com/zaccaria-lab/TP53loss_WGD]. / [https://doi.org/10.5281/zenodo.10658423].

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

## Acknowledgements

We would like to thank all the patients for their generosity in participating in the clinical trials and giving permission to share their data. We would like to thank The Francis CRICK Institute Science Technology Platforms for their continuing support; Experimental Histopathology, Biological Research Facility, Flow-Cytometry Facility, Advanced Sequencing, In Vivo Imaging, Light Microscopy STP, Bioinformatics & Biostatistics, Cell Services and High Throughput Screening; especially Mike Howell, Ming Jiang and Scott Warchal. We would like to acknowledge the RECIST working group for collecting the data in the EORTC database. This work was supported by the Francis Crick Institute that receives its core funding from Cancer Research UK (FC001169), the UK Medical Research Council (FC001169), and the Wellcome Trust (FC001169). For the purpose of Open Access, the author has applied a CC BY public copyright licence to any Author Accepted Manuscript version arising from this submission. L.A.H.'s work as Fellow at EORTC

Headquarters was supported by a grant from the Grierson Fund through the EORTC Cancer Research Fund (ECRF). S.Z. is a CRUK Career Development Fellow (award ref. RCCCDF-Nov21\100005) and is supported by the Rosetrees Trust (grant ref. M917). R.E.H. was supported by a Sir Henry Wellcome Postdoctoral Fellowship (Wellcome Trust; WT209199/Z/17). N.K. is supported by the Breast Cancer Research Foundation (BCRF 23-157), the Rosetrees Trust and Cancer Research UK. E.G. received support from the ERC Consolidator Grant THESEUS (grant agreement no.617844). B.B., W.H., and E.G. were funded by the ERC Advanced Grant PROTEUS, (grant agreement no. 835297). C.S. is a Royal Society Napier Research Professor (RSRP\R\210001). C.S. is funded by Cancer Research UK (TRACERx (C11496/A17786), PEACE (C416/A21999) and CRUK Cancer Immunotherapy Catalyst Network); Cancer Research UK Lung Cancer Centre of Excellence (C11496/A30025); the Rosetrees Trust, Butterfield and Stoneygate Trusts; NovoNordisk Foundation (ID16584); Royal Society Professorship Enhancement Award (RP/EA/180007); National Institute for Health Research (NIHR) University College London Hospitals Biomedical Research Centre; the Cancer Research UK-University College London Centre; Experimental Cancer Medicine Centre; the Breast Cancer Research Foundation (US); and The Mark Foundation for Cancer Research (Grant 21-029-ASP). This work was supported by a Stand Up To Cancer-LUNGevity-American Lung Association Lung Cancer Interception Dream Team Translational Research Grant (Grant Number: SU2C-AACR-DT23-17 to S.M. Dubinett and A.E. Spira). Stand Up To Cancer is a division of the Entertainment Industry Foundation. Research grants are administered by the American Association for Cancer Research, the Scientific Partner of SU2C. C.S. is in receipt of an ERC Advanced Grant (PROTEUS) from the European Research Council under the European Union's Horizon 2020 research and innovation program (grant agreement no. 835297).

## Author contributions

N.K., S.Z., E.G., and C.S. contributed equally to this work. S.H., N.K., and E.G. designed and undertook laboratory experiments. M.S., T.C., and C.T.H. analysed clinical data. B.B., H.v.d.B., and D.S. performed the single-cell sequencing supervised by F.F., M.A.B., A.M.F., T.B.K.W., B.B., A.H.R., W.W., G.K., K.L., and S.Z. carried out and/or interpreted bioinformatics analyses. D.O. and M.N. performed image cytometry analysis and data interpretation. M.J.R. developed Anafind. C.M.B., D.L.K., L.T., A.M., J.R.D., A.P.B., J.v.d.A., J.C., T.G.B., and C.B. provided patient samples and patient data. D.A.M. pathology assessment. S.V., W.H., A.H., C.M.R., J.R.M.B., M.A., and N.McG. carried out experiments or analysed bioinformatic data. L.A.H. and S.L. analysed and shared patient data. K.H.V. and J.D. gave critical comments on the manuscript. The TRACERx consortium coordinated the clinical trial, provided patient samples and patient data. C.S. supervised the project, and S.H., M.A.B., M.S., C.T.H., R.E.H., N.K., S.Z., and E.G. wrote the manuscript. S.H., S.K.C., E.C.d.B., C.T.H., E.G., and C.S. conceived and designed the project.

## Funding

## Competing interests

S.H. received a grant from AstraZeneca, M.A.B. has consulted for Achilles Therapeutics. C.T.H has received speaker fees from AstraZeneca and has a paid advisory role for GenesisCare UK, N.McG. has received consultancy fees and has stock options in Achilles Therapeutics; and holds European patents relating to targeting neoantigens (PCT/EP2016/059401), identifying patient response to immune checkpoint blockade (PCT/EP2016/071471), determining HLA LOH (PCT/GB2018/052004) and predicting survival rates of patients with cancer (PCT/GB2020/050221). K.L. has a patent (CA3068366A) on indel burden and CPI response pending and speaker fees from Roche

tissue diagnostics and Ellipses Pharmaceuticals, research funding from CRUK TDL/Ono/LifeArc alliance, Genesis Therapeutics and consulting roles with Monopteros Therapeutics and Kynos Therapeutics (all outside of this work). K.H.V. is on the board of directors and shareholder of Bristol Myers Squibb and on the scientific advisory board (with stock options) of PMV Pharma, RAZE Therapeutics, Volastra Pharmaceuticals and Kovina Therapeutics. She is on the scientific advisory board of Ludwig Cancer and a co-founder and consultant of Faeth Therapeutics. She has been in receipt of research funding from Astex Pharmaceuticals and AstraZeneca and contributed to CRUK Cancer Research Technology filing of patent application WO/2017/144877. T.G.B is supported by the NIH/NCI U54CA224081, R01CA169338, R01CA211052, R01CA204302, U01CA217882 and the Chan-Zuckerberg Biohub. N.K. acknowledges grants from AstraZeneca. C.S. acknowledges grants from AstraZeneca, Boehringer-Ingelheim, Bristol Myers Squibb, Pfizer, Roche-Ventana, Invitae (previously Archer Dx Inc - collaboration in minimal residual disease sequencing technologies), Ono Pharmaceutical, and Personalis. He is Chief Investigator for the AZ MeRmaiD 1 and 2 clinical trials and is the Steering Committee Chair. He is also Co-Chief Investigator of the NHS Galleri trial funded by GRAIL and a paid member of GRAIL's Scientific Advisory Board. He receives consultant fees from Achilles Therapeutics (also SAB member), Bicycle Therapeutics (also a SAB member), Genentech, Medicxi, China Innovation Centre of Roche (CICoR) formerly Roche Innovation Centre – Shanghai, Metabomed (until 30 July 2022), and the Sarah Cannon Research Institute C.S has received honoraria from Amgen, AstraZeneca, Bristol Myers Squibb, GlaxoSmithKline, Illumina, MSD, Novartis, Pfizer, and Roche-Ventana. C.S. has previously held stock options in Apogen Biotechnologies and GRAIL, and currently has stock options in Epic Bioscience, Bicycle Therapeutics, and has stock options and is co-founder of Achilles Therapeutics. C.S declares a patent application (PCT/US2017/028013) for methods to lung cancer); targeting neoantigens (PCT/EP2016/059401), identifying patent response to immune checkpoint blockade (PCT/EP2016/071471), determining HLA LOH (PCT/GB2018/052004); predicting survival rates of patients with cancer (PCT/GB2020/050221), identifying patients who respond to cancer treatment (PCT/GB2018/051912); methods for lung cancer detection (US20190106751A1). C.S. is an inventor on a European patent application (PCT/GB2017/053289) relating to assay technology to detect tumor recurrence. This patent has been licensed to a commercial entity and under their terms of employment C.S is due a revenue share of any revenue generated from such license(s). The remaining authors declare no competing interest.

## Additional information

[1]Cancer Evolution and Genome Instability Laboratory, The Francis Crick Institute, 1 Midland Rd, London NW1 1AT, UK. [2]Cancer Research UK Lung Cancer Centre of Excellence, University College London Cancer Institute, Paul O'Gorman Building, 72 Huntley Street, London WC1E 6BT, UK. [3]Department of Medical Oncology, University College London Hospitals, 235 Euston Rd, Fitzrovia, London NW1 2BU, UK. [4]Department of Oncology and Radiotherapy, Medical University of Gdańsk, ul. Mariana Smoluchowskiego 17, 80-214 Gdańsk, Poland. [5]European Research Institute for the Biology of Ageing, University of Groningen, University Medical Center Groningen, A. Deusinglaan 1, Groningen 9713, the Netherlands. [6]Oncology Data Science, Oncology R&D, AstraZeneca, Boston, MA, USA. [7]Late Development, Oncology R&D, AstraZeneca, Boston, MA, USA. [8]Research and Early Development, Oncology R&D, AstraZeneca, Cambridge, UK. [9]Research Department of Pathology, University College London Medical School, University Street, London WC1E 6JJ, UK. [10]Department of Medicine, University of California, San Francisco, CA 94158, USA. [11]Furlong Laboratory, EMBL Meyerhofstraße 1, 69117 Heidelberg, Germany. [12]European Organization for Research and Treatment of Cancer, Brussels, Belgium. [13]Bioinformatics & Biostatistics; Francis Crick Institute, London, UK. [14]Department of Cellular Pathology, University College London Hospitals, London, UK. [15]Advanced Light Microscopy, The Francis Crick Institute, 1 Midland Rd, London NW1 1AT, UK. [16]Cancer Genome Evolution Research Group, Cancer Research UK Lung Cancer Centre of Excellence, University College London Cancer Institute, London, UK. [17]Oncogene Biology Laboratory, The Francis Crick Institute, 1 Midland Rd, London NW1 1AT, UK. [18]p53 and Metabolism Laboratory, The Francis Crick Institute, 1 Midland Rd, London NW1 1AT, UK. [19]Chan-Zuckerberg Biohub, San Francisco, USA. [20]Computational Cancer Genomics Research Group, University College London Cancer Institute, London, UK. [122]These authors contributed equally: Sebastijan Hobor, Maise Al Bakir, Crispin T. Hiley, Marcin Skrzypski. [123]These authors jointly supervised this work: Nnennaya Kanu, Simone Zaccaria, Eva Grönroos, and Charles Swanton. ✉e-mail: n.kanu@ucl.ac.uk; s.zaccaria@ucl.ac.uk; Eva.Gronroos@crick.ac.uk; Charles.Swanton@crick.ac.uk

## TRACERx consortium

Charles Swanton [1,2,3,123]✉, Eva Grönroos [1,123]✉, Maise Al Bakir [1,122], Crispin T. Hiley[1,2,3,122], Alexander M. Frankell [1,2], Thomas B. K. Watkins[1], David A. Moore [2,14], William Hill[1], Ariana Huebner [1,2,16], Carlos Martínez-Ruiz [2,16], Mihaela Angelova [1], Nicholas McGranahan [2,16], Kevin Litchfield[1], Robert E. Hynds [1,2], Nnennaya Kanu[2,123]✉, Simone Zaccaria [2,20,123]✉, Jason F. Lester[21], Amrita Bajaj[22], Apostolos Nakas[22], Azmina Sodha-Ramdeen[22], Mohamad Tufail[22], Molly Scotland[22], Rebecca Boyles[22], Sridhar Rathinam[22], Claire Wilson[23], Domenic Marrone[24], Sean Dulloo[22,24], Dean A. Fennell[22,24], Gurdeep Matharu[25], Jacqui A. Shaw[25], Ekaterini Boleti[26], Heather Cheyne[27], Mohammed Khalil[27], Shirley Richardson[27], Tracey Cruickshank[27], Gillian Price[28,29], Keith M. Kerr[29,30], Sarah Benafif[3,31], Jack French[31], Kayleigh Gilbert[31], Babu Naidu[32], Akshay J. Patel[33], Aya Osman[34], Carol Enstone[34], Gerald Langman[34], Helen Shackleford[34], Madava Djearaman[34], Salma Kadiri[34], Gary Middleton[34,35], Angela Leek[36], Jack Davies Hodgkinson[36], Nicola Totton[36], Angeles Montero[37], Elaine Smith[37], Eustace Fontaine[37], Felice Granato[37], Antonio Paiva-Correia[38], Juliette Novasio[37], Kendadai Rammohan[37], Leena Joseph[37], Paul Bishop[37], Rajesh Shah[37], Stuart Moss[37], Vijay Joshi[37], Philip A. J. Crosbie[37,39,40], Katherine D. Brown[40,41], Mathew Carter[40,41], Anshuman Chaturvedi[40,41], Pedro Oliveira[40,41], Colin R. Lindsay[40,42], Fiona H. Blackhall[40,42], Matthew G. Krebs[42], Yvonne Summers[40,42], Alexandra Clipson[40,43], Jonathan Tugwood[40,43], Alastair Kerr[40,43], Dominic G. Rothwell[40,43], Caroline Dive[40,43], Hugo J. W. L. Aerts[44,45,46], Roland F. Schwarz[47,48], Tom L. Kaufmann[48,49], Gareth A. Wilson[1], Rachel Rosenthal[1], Peter Van Loo[50,51,52], Nicolai J. Birkbak[1,2,53,54,55], Zoltan Szallasi[56,57,58], Judit Kisistok[53,54,55], Mateo Sokac[53,54,55], Roberto Salgado[59,60], Miklos Diossy[56,57,61], Jonas Demeulemeester[62,63,64], Abigail Bunkum[2,20,65], Angela Dwornik[66], Alastair Magness[67], Andrew J. Rowan[1], Angeliki Karamani[66], Antonia Toncheva[2], Benny Chain[66], Carla Castignani[52,68], Chris Bailey[1], Christopher Abbosh[2], Clare Puttick[1,2,16], Clare E. Weeden[67], Claudia Lee[1], Corentin Richard[2], Cristina Naceur-Lombardelli[2], David R. Pearce[66], Despoina Karagianni[66], Dhruva Biswas[1,2,69], Dina Levi[67], Elizabeth Larose Cadieux[52,68], Emilia L. Lim[1,2], Emma Colliver[1], Emma Nye[70], Felip Gálvez-Cancino[66], Francisco Gimeno-Valiente[2], George Kassiotis[67,71], Georgia Stavrou[66], Gerasimos-Theodoros Mastrokalos[66], Helen L. Lowe[66], Ignacio Garcia Matos[66], Imran Noorani[67], Jacki Goldman[67], James L. Reading[66], James R. M. Black [2,16],

Jayant K. Rane[1,66], Jerome Nicod[72], John A. Hartley[66], Karl S. Peggs[73,74], Katey S. S. Enfield[1], Kayalvizhi Selvaraju[66], Kerstin Thol[2,16], Kevin W. Ng[75], Kezhong Chen[66], Krijn Dijkstra[67], Kristiana Grigoriadis[1,2,16], Krupa Thakkar[2], Leah Ensell[66], Mansi Shah[66], Maria Litovchenko[66], Mariam Jamal-Hanjani[2,65,76], Mariana Werner Sunderland[2], Matthew R. Huska[77], Mark S. Hill[1], Michelle Dietzen[1,2,16], Michelle M. Leung[1,2,16], Mickael Escudero[67], Miljana Tanić[68,78], Monica Sivakumar[2], Olga Chervova[66,79], Olivia Lucas[1,2,20,80], Oriol Pich[1], Othman Al-Sawaf[66,81], Paulina Prymas[2], Philip Hobson[67], Piotr Pawlik[66], Richard Kevin Stone[70], Robert Bentham[2,16], Roberto Vendramin[1,2,82], Sadegh Saghafinia[2], Samuel Gamble[66], Selvaraju Veeriah[2], Seng Kuong Anakin Ung[66], Sergio A. Quezada[2,83], Sharon Vanloo[2], Sonya Hessey[2,20,65], Sophia Ward[1,2,72], Sian Harries[1,2,72], Stefan Boeing[67], Stephan Beck[68], Supreet Kaur Bola[66], Takahiro Karasaki[1,2,65], Tamara Denner[67], Teresa Marafioti[14], Thomas Patrick Jones[16], Victoria Spanswick[66], Vittorio Barbè[67], Wei-Ting Lu[67], Wing Kin Liu[2,65], Yin Wu[66], Yutaka Naito[67], Zoe Ramsden[67], Catarina Veiga[84], Gary Royle[85], Charles-Antoine Collins-Fekete[86], Francesco Fraioli[87], Paul Ashford[88], Martin D. Forster[2,3], Siow Ming Lee[2,3], Elaine Borg[14], Mary Falzon[14], Dionysis Papadatos-Pastos[3], James Wilson[3], Tanya Ahmad[3], Alexander James Procter[89], Asia Ahmed[89], Magali N. Taylor[89], Arjun Nair[89,90], David Lawrence[91], Davide Patrini[91], Neal Navani[92,93], Ricky M. Thakrar[92,93], Sam M. Janes[94], Emilie Martinoni Hoogenboom[80], Fleur Monk[80], James W. Holding[80], Junaid Choudhary[80], Kunal Bhakhri[80], Marco Scarci[80], Pat Gorman[80], Reena Khiroya[14], Robert C. M. Stephens[80], Yien Ning Sophia Wong[80], Zoltan Kaplar[95,96], Steve Bandula[80], Allan Hackshaw[97], Anne-Marie Hacker[97], Abigail Sharp[97], Sean Smith[97], Harjot Kaur Dhanda[97], Camilla Pilotti[97], Rachel Leslie[97], Anca Grapa[98], Hanyun Zhang[98], Khalid AbdulJabbar[99], Xiaoxi Pan[100], Yinyin Yuan[100], David Chuter[101], Mairead MacKenzie[101], Serena Chee[102], Aiman Alzetani[102], Judith Cave[103], Jennifer Richards[102], Eric Lim[104,105], Paulo De Sousa[105], Simon Jordan[105], Alexandra Rice[105], Hilgardt Raubenheimer[105], Harshil Bhayani[105], Lyn Ambrose[105], Anand Devaraj[105], Hema Chavan[105], Sofina Begum[105], Silviu I. Buderi[105], Daniel Kaniu[105], Mpho Malima[105], Sarah Booth[105], Andrew G. Nicholson[105,106], Nadia Fernandes[105], Pratibha Shah[105], Chiara Proli[105], Madeleine Hewish[107,108], Sarah Danson[109,110], Michael J. Shackcloth[111], Lily Robinson[112], Peter Russell[112], Kevin G. Blyth[113,114,115], Andrew Kidd[116], Craig Dick[117], John Le Quesne[118,119,120], Alan Kirk[121], Mo Asif[121], Rocco Bilancia[121], Nikos Kostoulas[121] & Mathew Thomas[121]

[21]Singleton Hospital, Swansea Bay University Health Board, Swansea, UK. [22]University Hospitals of Leicester NHS Trust, Leicester, UK. [23]Leicester Medical School, University of Leicester, Leicester, UK. [24]University of Leicester, Leicester, UK. [25]Cancer Research Centre, University of Leicester, Leicester, UK. [26]Royal Free London NHS Foundation Trust, London, UK. [27]Aberdeen Royal Infirmary NHS Grampian, Aberdeen, UK. [28]Department of Medical Oncology, Aberdeen Royal Infirmary NHS Grampian, Aberdeen, UK. [29]University of Aberdeen, Aberdeen, UK. [30]Department of Pathology, Aberdeen Royal Infirmary NHS Grampian, Aberdeen, UK. [31]The Whittington Hospital NHS Trust, London, UK. [32]Birmingham Acute Care Research Group, Institute of Inflammation and Ageing, University of Birmingham, Birmingham, UK. [33]Guy's and St Thomas' NHS Foundation Trust, London, UK. [34]University Hospital Birmingham NHS Foundation Trust, Birmingham, UK. [35]Institute of Immunology and Immunotherapy, University of Birmingham, Birmingham, UK. [36]Manchester Cancer Research Centre Biobank, Manchester, UK. [37]Wythenshawe Hospital, Manchester University NHS Foundation Trust, Manchester, UK. [38]Manchester University NHS Foundation Trust, Manchester, UK. [39]Division of Infection, Immunity and Respiratory Medicine, University of Manchester, Manchester, UK. [40]Cancer Research UK Lung Cancer Centre of Excellence, University of Manchester, Manchester, UK. [41]The Christie NHS Foundation Trust, Manchester, UK. [42]Division of Cancer Sciences, The University of Manchester and The Christie NHS Foundation Trust, Manchester, UK. [43]Cancer Research UK Manchester Institute Cancer Biomarker Centre, University of Manchester, Manchester, UK. [44]Artificial Intelligence in Medicine (AIM) Program, Mass General Brigham, Harvard Medical School, Boston, MA, USA. [45]Department of Radiation Oncology, Brigham and Women's Hospital, Dana-Farber Cancer Institute, Harvard Medical School, Boston, MA, USA. [46]Radiology and Nuclear Medicine, CARIM & GROW, Maastricht University, Maastricht, The Netherlands. [47]Institute for Computational Cancer Biology, Center for Integrated Oncology (CIO), Cancer Research Center Cologne Essen (CCCE), Faculty of Medicine and University Hospital Cologne, University of Cologne, Cologne, Germany. [48]Berlin Institute for the Foundations of Learning and Data (BIFOLD), Berlin, Germany. [49]Berlin Institute for Medical Systems Biology, Max Delbrück Center for Molecular Medicine in the Helmholtz Association (MDC), Berlin, Germany. [50]Department of Genetics, The University of Texas MD Anderson Cancer Center, Houston, Texas, USA. [51]Department of Genomic Medicine, The University of Texas MD Anderson Cancer Center, Houston, Texas, USA. [52]Cancer Genomics Laboratory, The Francis Crick Institute, London, UK. [53]Department of Molecular Medicine, Aarhus University Hospital, Aarhus, Denmark. [54]Department of Clinical Medicine, Aarhus University, Aarhus, Denmark. [55]Bioinformatics Research Centre, Aarhus University, Aarhus, Denmark. [56]Danish Cancer Society Research Center, Copenhagen, Denmark. [57]Computational Health Informatics Program, Boston Children's Hospital, Boston, MA, USA. [58]Department of Bioinformatics, Semmelweis University, Budapest, Hungary. [59]Department of Pathology, ZAS Hospitals, Antwerp, Belgium. [60]Division of Research, Peter MacCallum Cancer Centre, Melbourne, Australia. [61]Department of Physics of Complex Systems, ELTE Eötvös Loránd University, Budapest, Hungary. [62]Integrative Cancer Genomics Laboratory, VIB Center for Cancer Biology, Leuven, Belgium. [63]VIB Center for AI & Computational Biology, Leuven, Belgium. [64]Department of Oncology, KU Leuven, Leuven, Belgium. [65]Cancer Metastasis Laboratory, University College London Cancer Institute, London, UK. [66]University College London Cancer Institute, London, UK. [67]The Francis Crick Institute, London, UK. [68]Medical Genomics, University College London Cancer Institute, London, UK. [69]Bill Lyons Informatics Centre, University College London Cancer Institute, London, UK. [70]Experimental Histopathology, The Francis Crick Institute, London, UK. [71]Department of Infectious Disease, Faculty of Medicine, Imperial College London, London, UK. [72]Advanced Sequencing Facility, The Francis Crick Institute, London, UK. [73]Department of Haematology, University College London Hospitals, London, UK. [74]Cancer Immunology Unit, Research Department of Haematology, University College London Cancer Institute, London, UK. [75]Retroviral Immunology Group, The Francis Crick Institute, London, UK. [76]Department of Medical Oncology, University College London Hospitals, London, UK. [77]Bioinformatics and Systems Biology, Method Development and Research Infrastructure, Robert Koch Institute, Nordufer 20, 13353 Berlin, Germany. [78]Experimental Oncology, Institute for Oncology and Radiology of Serbia, Belgrade, Serbia. [79]University College London Department of Epidemiology and Health Care, London, UK. [80]University College London Hospitals, London, UK. [81]Department I of Internal Medicine, University Hospital of Cologne, Cologne, Germany. [82]Tumour Immunogenomics and Immunosurveillance Laboratory, University College London Cancer Institute, London, UK. [83]Immune

Regulation and Tumour Immunotherapy Group, Cancer Immunology Unit, Research Department of Haematology, University College London Cancer Institute, London, UK. [84]Centre for Medical Image Computing, Department of Medical Physics and Biomedical Engineering, London, UK. [85]Department of Medical Physics and Bioengineering, University College London Cancer Institute, London, UK. [86]Department of Medical Physics and Biomedical Engineering, University College London, London, UK. [87]Institute of Nuclear Medicine, Division of Medicine, University College London, London, UK. [88]Institute of Structural and Molecular Biology, University College London, London, UK. [89]Department of Radiology, University College London Hospitals, London, UK. [90]UCL Respiratory, Department of Medicine, University College London, London, UK. [91]Department of Thoracic Surgery, University College London Hospital NHS Trust, London, UK. [92]Lungs for Living Research Centre, UCL Respiratory, University College London, London, UK. [93]Department of Thoracic Medicine, University College London Hospitals, London, UK. [94]Lungs for Living Research Centre, UCL Respiratory, Department of Medicine, University College London, London, UK. [95]Integrated Radiology Department, North-buda St. John's Central Hospital, Budapest, Hungary. [96]Institute of Nuclear Medicine, University College London Hospitals, London, UK. [97]Cancer Research UK & UCL Cancer Trials Centre, London, UK. [98]The Institute of Cancer Research, London, UK. [99]Case45, London, UK. [100]The University of Texas MD Anderson Cancer Center, Houston, Texas, USA. [101]Independent Cancer Patient's voice, London, UK. [102]University Hospital Southampton NHS Foundation Trust, Southampton, UK. [103]Department of Oncology, University Hospital Southampton NHS Foundation Trust, Southampton, UK. [104]Academic Division of Thoracic Surgery, Imperial College London, London, UK. [105]Royal Brompton and Harefield Hospitals, part of Guy's and St Thomas' NHS Foundation Trust, London, UK. [106]National Heart and Lung Institute, Imperial College, London, UK. [107]Royal Surrey Hospital, Royal Surrey Hospitals NHS Foundation Trust, Guildford, UK. [108]University of Surrey, Guildford, UK. [109]University of Sheffield, Sheffield, UK. [110]Sheffield Teaching Hospitals NHS Foundation Trust, Sheffield, UK. [111]Liverpool Heart and Chest Hospital, Liverpool, UK. [112]Princess Alexandra Hospital, The Princess Alexandra Hospital NHS Trust, Harlow, UK. [113]School of Cancer Sciences, University of Glasgow, Glasgow, UK. [114]Beatson Institute for Cancer Research, University of Glasgow, Glasgow, UK. [115]Queen Elizabeth University Hospital, Glasgow, UK. [116]Institute of Infection, Immunity & Inflammation, University of Glasgow, Glasgow, UK. [117]NHS Greater Glasgow and Clyde, Glasgow, UK. [118]Cancer Research UK Scotland Institute, Glasgow, UK. [119]Institute of Cancer Sciences, University of Glasgow, Glasgow, UK. [120]NHS Greater Glasgow and Clyde Pathology Department, Queen Elizabeth University Hospital, Glasgow, UK. [121]Golden Jubilee National Hospital, Clydebank, UK.

