## [Peer Review File · Nature Communications]

Mixed responses to targeted therapy driven by chromosomal instability through p53 dysfunction and genome doublingThis manuscript has been previously reviewed at another journal that is not operating a transparent peer review scheme. This document only contains reviewer comments and rebuttal letters for versions considered at *Nature Communications*.

REVIEWER COMMENTS

Reviewer #1 (Remarks to the Author):

The authors have provided reasonable responses to the majority of the reviewer criticisms.

Reviewer #4 (Remarks to the Author):

The study by Hobor et al. investigates the molecular basis of heterogeneous treatment responses of different tumor nodules within the same patient/mouse with EGFR mutated NSCLC with and without p53 alterations (EP and E). Reanalysis of clinical data showed an association between p53 loss and a higher level of intra-patient heterogeneity regarding TKI response. These findings were validated in a mouse model which was further used to characterize the genomic and clonal architecture of E and EP tumors. In tissue culture experiments, cells were sorted for ploidy and the team showed that whole genome doubling (WGD) on the p53 deficient background is a critical factor to create and tolerate a high level genomic instability which in turn is likely to influence treatment response.

The study is original and adds significant new aspects to the field as it provides a genomic explanation for a clinical phenomenon. It is conducted at a high technical level, has a logical flow, refers to the relevant literature and is well written. It provides to my knowledge for the first time quantifiable evidence for the intra-patient heterogeneity on treatment response of EGFR LUAD stratified for TP53 mutations and the genomic dynamics of TP53 mutated lung cancer in the context of TKI resistance.

An earlier version of the manuscript obviously has been reviewed in the context of another journal. In the provided point by point response, the authors have addressed all raised points.

I have the following points that should be considered.

Major points

- 1) Given the limited number of mice included in the SCNA analysis, the differences in SCNAs between E and EP should not be over-interpreted as the trend for the human SCNAs usually goes in the same directions for E and EP (Fig. 2A). E.g. highlighting CDKN2A as exclusively lost in E tumors while PTEN loss is only observed in EP tumors seems to be phrased too strong since the human SCNA profiles show similar albeit weaker trends in the respective other groups.
- 2) Regarding the siRNA screen: were all 43 genes gained in all hexaploid subclones or was there heterogeneity which could explain the heterogeneous responses in the screen? The results of the siRNA screen are quite inconsistent and difficult to interpret. For most siRNAs, there are other clones where the knock down in control conditions also reduces the viability. For BCL9 for example, clone 1 (top) shows reduced viability only under control conditions (without erlotinib), clone 3 shows reduced viability for control and erlotinib and only clone 4 shows selectively reduced viability under erlotinib treatment. The knock down of a number of genes shows reduced viability for erlotinib for one clone but not the other three. Has a positive control e.g. against EGFR been used? Have the experiments been repeated (for a few genes) with the same results for the same clones? The data does not allow the conclusion in line 591 "... indicating that copy number gains in these genes conferred EGFR TKI resistance." Also in the discussion in line 645 "... highlighting the diversity of resistance mechanism as a result of SCNAs." I am not sure whether SCNAs or the lack of stability of the assay are the reason for the heterogeneous results.

Minor points

- 1) Why has the TCGA dataset been excluded from Extended Fig. 5?
- 2) The phylogenetic tree in Ext. Data Fig. 6D needs more explanation. The x-axis length of the lines is proportional to the number or fraction of the genome altered relative to the MCRA? Some of the symbols are not aligned to the lines of the tree. What does it mean? Do the grey ends represent dead ends? If yes, no clade is alive? Is the x-axis considered proportional to clonal viability where an early structural change is interpreted as beneficial if many subsequent clades build on it?
- 3) According to Fig. 4C, there are 18 out of 40 resistant hexaploid clones with 19 mutations where one clone has an EGFR T790M and CIC R274X mutation. The legend of Fig. 4C is in disagreement with Fig. 4C regarding the description of the ploidy on the top (ploidy vs. parental ploidy). I

assume, the figure panel is correct.

4) The median absolute deviation (MAD) of tumor sizes is displayed in two different ways, as volume fold change (0 to 50) in Fig. 2C and Ext Data Fig. 3F while it is % diameter change from base line (-100 to +200) in Fig. 1F and Ext. Data Fig. 3A. This might be harmonized.

RESPONSE TO REVIEWERS' COMMENTS

Text changes in the manuscript are highlighted in yellow both in the manuscript and here to assist the reviewer.

REVIEWER COMMENTS:

Reviewer #1 (Remarks to the Author):

The authors have provided reasonable responses to the majority of the reviewer criticisms.

Reviewer #4 (Remarks to the Author):

The study by Hobor et al. investigates the molecular basis of heterogeneous treatment responses of different tumor nodules within the same patient/mouse with EGFR mutated NSCLC with and without p53 alterations (EP and E). Reanalysis of clinical data showed an association between p53 loss and a higher level of intra-patient heterogeneity regarding TKI response. These findings were validated in a mouse model which was further used to characterize the genomic and clonal architecture of E and EP tumors. In tissue culture experiments, cells were sorted for ploidy and the team showed that whole genome doubling (WGD) on the p53 deficient background is a critical factor to create and tolerate a high level genomic instability which in turn is likely to influence treatment response. The study is original and adds significant new aspects to the field as it provides a genomic explanation for a clinical phenomenon. It is conducted at a high technical level, has a logical flow, refers to the relevant literature and is well written. It provides to my knowledge for the first time quantifiable evidence for the intra-patient heterogeneity on treatment response of EGFR LUAD stratified for TP53 mutations and the genomic dynamics of TP53 mutated lung cancer in the context of TKI resistance.

An earlier version of the manuscript obviously has been reviewed in the context of another journal. In the provided point by point response, the authors have addressed all raised points. I have the following points that should be considered.

Major points

1) Given the limited number of mice included in the SCNA analysis, the differences in SCNAs between E and EP should not be over-interpreted as the trend for the human SCNAs usually goes in the same directions for E and EP (Fig. 2A). E.g. highlighting CDKN2A as exclusively lost in E tumors while PTEN loss is only observed in EP tumors seems to be phrased too strong since the human SCNA profiles show similar albeit weaker trends in the respective other groups.

We thank the reviewer for this comment and have amended the paragraph to read:

We evaluated the suitability of GEMMs as a model system to assess heterogeneous responses in EGFR-driven lung cancer by generating combined synteny SCNA maps of treatment naïve mouse and human tumors. Re-mapping the mouse LUAD genome onto the human LUAD genome revealed that, in all samples investigated, oncogenes, such as *AKT1* and tumor suppressors, such as *PBRM1*, *SETD2*, *BAP1*, and *SMAD3*, were recurrently affected by copy-number gains (pink) and losses (blue) respectively, in both human and mouse tumours irrespective of E or EP status (Fig. 2A). In contrast syntenic gains or losses that were restricted to either E or EP tumors included the tumor suppressor gene *CDKN2A*, which was predominantly lost in mouse and human E tumors. Whereas syntenic loss

of *PTEN*, which has been associated with TKI resistance³⁴, was mainly observed in treatment naïve EP tumors. These analyses demonstrate that tumors from E and EP mice recapitulate several of the genomic events observed in human tumors and highlight the potential importance of a limited set of genes commonly gained or lost in the earliest stages of *EGFR*-driven tumorigenesis (Fig. 2A, Extended Data Table 3).

We hope that this change highlights our intention to demonstrate that the *EGFR* model used sufficiently recapitulates the genomic alterations occurring in patient with *EGFR*-mutated tumours with or without *TP53* loss. As the murine tumours analysed are in the mm range as opposed to patients with tumours which would be larger than 1 cm, we acknowledge that the murine model system fails to fully recapitulate the full extent of aberrations that will have occurred during the progression of the human disease. The modifications to the text reflect our emphasis of the syntenic changes present in both species with *EGFR* driven lung cancers.

2) Regarding the siRNA screen: were all 43 genes gained in all hexaploid subclones or was there heterogeneity which could explain the heterogeneous responses in the screen?

We observed heterogeneity in the copy number alterations in the hexaploid resistant subclones and not all 43 genes were gained in all the 4 subclones used in the siRNA screen. We agree this heterogeneity is a likely cause of the heterogeneous responses observed in the screen. We apologise for the lack of clarity as this is the concept we were trying to convey. That with heterogenous subclones, with heterogenous copy number alterations, there is variable sensitivity to single gene targeting approaches and resistance can evolve via different pathways in different subclones. We also believe that some of the gained genes may be co amplified passenger genes and may not be essential for the resistance to erlotinib hence why siRNA silencing does not impact cell viability. We have now included a supplementary figure displaying the individual genes and the extent of their gains in each subclone. We have also included a sentence in the text to clarify the presence of heterogeneity in copy number gains in the individual subclones:

“To validate novel copy-number mechanisms of erlotinib resistance in PC9 cells, a functional siRNA screen was performed in four hexaploid resistant subclones with distinct copy number changes (Extended Data Fig. 10C).”

The results of the siRNA screen are quite inconsistent and difficult to interpret. For most siRNAs, there are other clones where the knock down in control conditions also reduces the viability. For *BCL9* for example, clone 1 (top) shows reduced viability only under control conditions (without erlotinib), clone 3 shows reduced viability for control and erlotinib and only clone 4 shows selectively reduced viability under erlotinib treatment.

We apologise for the lack of clarity in this section. All hexaploid subclones used in the screen were generated by long term culture under erlotinib selection. Since these subclones are resistant to erlotinib, we expect that the subclones may have become dependent on some of the gains to different extents. As such, we expect some loss of viability following depletion of some of these siRNAs to varying extents in the absence of erlotinib. Please see Extended Data Fig 10C which highlights the copy number state of the parental hexaploid clones.

The knock down of a number of genes shows reduced viability for erlotinib for one clone but not the other three.

We apologise for our lack of clarity in explaining our expectation and interpretation of these results. The four resistant hexaploid subclones used in the screen had distinct CNAs (See Extended Data Figure 10C). Due to the heterogeneity of copy number aberrations between the subclones, we do expect variability in the response of individual subclones to each siRNA. We intended to convey to the reader that with heterogenous subclones, derived from a common precursor, the presences of heterogenous copy number alterations will result in variable sensitivity to single gene targeting approaches. In summary, resistance can evolve via different pathways in different subclones.

Has a positive control e.g. against EGFR been used?

The readout for the screen was cell viability as quantified using Dapi staining. Accordingly, the positive control we used for the screen was a universally lethal control siRNA against ubiquitin B (siUBB).

Have the experiments been repeated (for a few genes) with the same results for the same clones?

For each clone, three independent transfections were performed for each siRNA. For the majority of the genes the results were highly concordant. We apologize for the lack of clarity in the methods section and have updated it to include this information.

“The screen was performed in triplicate and UBB was used as a positive control to assess loss of viability. Non-targeting controls were used for each parental and daughter clone to establish base line conditions.”

The data does not allow the conclusion in line 591 “... indicating that copy number gains in these genes conferred EGFR TKI resistance.” Also in the discussion in line 645 “... highlighting the diversity of resistance mechanism as a result of SCNAs.” I am not sure whether SCNAs or the lack of stability of the assay are the reason for the heterogeneous results.

We apologise for the lack of clarity. All replicates were highly concordant in the siRNA cell viability screen. Within the copy number gains identified after erlotinib treatment of the hexaploid subclones priority was given to genes previously implicated in TKI resistance. Our data demonstrate there is heterogeneity in the copy number gains in the different resistant subclones which may reflect the existence of different routes to resistance in different subclones.

We have accordingly toned down our text to say

Erlotinib resistance was assessed by comparing cell numbers following gene silencing in the presence or absence of erlotinib (see Online Methods, Fig. 4f). In total, of the 43 gained genes investigated, 14, including NRAS, ERBB3, HRAS, BRAF and PDGFR β led to EGFR TKI re-sensitisation (FDR<0.1, light blue triangles) in at least 1 hexaploid subclone after siRNA mediated knockdown (Fig. 4F) suggesting that, depending on the subclone, copy number gains in these genes might contribute to TKI resistance and that, in line with reported results⁴⁹ different resistance mechanisms might be adopted by the individual resistant subclones.

Minor points

1) Why has the TCGA dataset been excluded from Extended Fig. 5?

We thank the reviewer for pointing out this omission and have included an analysis of TMB within the EGFR and EGFR p53 TCGA lung adenocarcinoma data set. We have now included this dataset as Extended Figure 5e. The Figure legend has been updated to reflect this addition and now reads:

E, Dot plot showing a non-significant difference in mutation burden between patients with E (n=19, yellow) or EP (n=38, turquoise) tumors in the TCGA cohort ($p=0.4775$, Student's exact test).

2) The phylogenetic tree in Ext. Data Fig. 6D needs more explanation. The x-axis length of the lines is proportional to the number or fraction of the genome altered relative to the MCRA? Some of the symbols are not aligned to the lines of the tree. What does it mean? Do the grey ends represent dead ends? If yes, no clade is alive? Is the x-axis considered proportional to clonal viability where an early structural change is interpreted as beneficial if many subsequent clades build on it?

We profusely and sincerely thank the reviewer for pointing out the lack of clarity and for spotting the PDF conversion error resulting in misplaced symbols. We have rectified this issue and added a corrected version of the phylogenetic tree. We have also extended the figure legend to:

Extended Data Fig. 6D, A phylogenetic tree where each terminal node (leaf, grey circle) represents a single cell from a murine lung tumour and genetic 'relatedness' is inferred using shared somatic copy number alterations (SCNAs) determined using single cell shallow WGS and MEDICC (see methods). Internal nodes of the tree are inferred by MEDICC. The length of each edge on the tree represents the number of SCNA events that have occurred during the transition from the parental to the daughter node in the phylogeny. These events are also represented by coloured squares and circles to the left of each edge with the effected chromosome denoted.

D

3) According to Fig. 4C, there are 18 out of 40 resistant hexaploid clones with 19 mutations where one clone has an EGFR T790M and CIC R274X mutation. The legend of Fig. 4C is in disagreement with Fig. 4C regarding the description of the ploidy on the top (ploidy vs. parental ploidy). I assume, the figure panel is correct.

The reviewer is correct and we apologize for the lack of clarity in the figure legend. We have now corrected the legend of Fig. 4C as reported here below:

C, (Left panel) Presence of somatic mutations in genes related to the EGFR pathway (black squares) is reported across all triploid (upper blue row) and hexaploid (upper red row) resistant daughter clones derived from either triploid (lower blue row) or hexaploid (lower red row) parental clones. (Right panel) Ploidy-relative copy-number gains (red colours) are reported for resistant daughter clones that have changed their ploidy state for 13 genes whose gain is known to have a role on TKI resistance.

4) The median absolute deviation (MAD) of tumor sizes is displayed in two different ways, as volume fold change (0 to 50) in Fig. 2C and Ext Data Fig. 3F while it is % diameter change from base line (-100 to +200) in Fig. 1F and Ext. Data Fig. 3A. This might be harmonized.

We thank the reviewer for this comment and have now remeasured the diameter for the entirety of the murine dataset with over 750 measurements as the data could not be extracted from the volumetric analysis. All graphs now show the percentage diameter change from baseline for each tumour lesion. The Extended figure legend 3 and figure legend 2 have been updated to clarify this.

REVIEWERS' COMMENTS

Reviewer #4 (Remarks to the Author):

The authors have addressed all points I raised.

Reviewer #4 (Remarks on code availability):

I just had a brief look at the github documentation. This is not my area of expertise. It looks fine to me.